# Cryo-EM visualization of the ribosome in termination complex with apo-RF3 and RF1

Jesper Pallesen[1], Yaser Hashem[1], Gürkan Korkmaz[2], Ravi Kiran Koripella[2], Chenhui Huang[2], Måns Ehrenberg[2], Suparna Sanyal[2], Joachim Frank[1,3]*

[1]Department of Biochemistry and Molecular Biophysics, Howard Hughes Medical Institute, Columbia University, New York City, United States; [2]Department of Cell and Molecular Biology, Uppsala University, Uppsala, Sweden; [3]Department of Biological Sciences, Columbia University, New York City, United States

**Abstract** Termination of messenger RNA translation in Bacteria and Archaea is initiated by release factors (RFs) 1 or 2 recognizing a stop codon in the ribosomal A site and releasing the peptide from the P-site transfer RNA. After release, RF-dissociation is facilitated by the G-protein RF3. Structures of ribosomal complexes with RF1 or RF2 alone or with RF3 alone—RF3 bound to a non-hydrolyzable GTP-analog—have been reported. Here, we present the cryo-EM structure of a post-termination ribosome containing both apo-RF3 and RF1. The conformation of RF3 is distinct from those of free RF3•GDP and ribosome-bound RF3•GDP(C/N)P. Furthermore, the conformation of RF1 differs from those observed in RF3-lacking ribosomal complexes. Our study provides structural keys to the mechanism of guanine nucleotide exchange on RF3 and to an L12-mediated ribosomal recruitment of RF3. In conjunction with previous observations, our data provide the foundation to structurally characterize the complete action cycle of the G-protein RF3.

*For correspondence: jf2192@columbia.edu

**Reviewing editor**: John Kuriyan, University of California, Berkeley, United States

## Introduction

Termination of bacterial messenger RNA (mRNA) translation depends on recognition of any one of the three stop codons in the ribosomal A site by either class-1 release factor 1 (RF1), recognizing UAA and UAG, or 2 (RF2), recognizing UAA and UGA. RF1 and RF2 induce release of the polypeptide attached to the P-site transfer RNA (tRNA) (for reviews, see *Kisselev and Buckingham, 2000*; *Klaholz, 2011*). Subsequent release of the class-1 RF from the post-polypeptide release complex (RC) is facilitated by a class-2 release factor, the G-protein RF3, in a GTP-dependent manner (*Freistroffer et al., 1997*). RF3, along with initiation factor 2 (IF2), elongation factor Tu (EF-Tu) and elongation factor G (EF-G), constitute a set of four major translational GTPases (*Zavialov and Ehrenberg, 2003*). Free RF3 has high affinity to GDP; the RF3•GDP complex dissociates very slowly into RF3 and GDP and, thus, free RF3 in the cytoplasm exists mainly in the GDP-form (*Zavialov et al., 2001, 2002*). In contrast to EF-Tu, for which GDP exchange to GTP is conferred by the guanine nucleotide exchange factor EF-Ts off the ribosome, the guanine nucleotide exchange factor for RF3 is the ribosome in complex with a class-1 RF (*Zavialov et al., 2001*). GDP-to-GTP exchange occurs rapidly on free IF2 and EF-G in the absence of any exchange factor. As RF3•GDP enters the class-1 RF-bound ribosome, GDP is readily exchanged for GTP, provided the peptide has been hydrolyzed off the P-site tRNA (*Zavialov et al., 2002*). GTP binding is accompanied by conformational changes in both RF3 and the ribosome (*Gao et al., 2007*). The change in ribosome structure promotes rapid release of the class-1 RF followed by GTP hydrolysis and, finally, release of RF3 in the GDP form from the ribosome (*Zavialov et al., 2001, 2002*).

**eLife digest** Ribosomes are complex molecular machines that join amino acids together to form proteins. The order of amino acids in the protein is specified by a strand of messenger RNA (mRNA), and the process of decoding the mRNA into a string of amino acids is called translation. A ribosome consists of two subunits—one large, one small—that come together at a particular site on the mRNA strand called the translation initiation site. The ribosome then moves along the mRNA—joining together amino acids brought to it by transfer RNA (tRNA)—until it reaches a termination site and releases the protein.

The ribosome has three sites; the first amino acid to be delivered by a tRNA molecule to the ribosome occupies the site in the middle—also called the P site—and the second amino acid is delivered to the A site. Once the first two amino acids have been joined together, the ribosome moves along the mRNA so that the first amino acid now occupies the third site, called the E or exit site, and the second amino acid occupies the P site, leaving the A site vacant. The third amino acid is then delivered to the A site, and the whole process repeats itself until the ribosome reaches the termination site. Proteins called release factors are responsible for terminating the translation process and releasing the translated string of amino acids, which folds to form a protein. In bacteria this task can by performed by two releases factors, known as RF1 and RF2. However, the release factor must itself be released to leave the ribosome free to translate another strand of mRNA.

Pallesen et al. have used cryo-electron microscopy (cryo-EM) to study how a third release factor, RF3, helps to release RF1 from the ribosome in bacteria. In cells, RF3 usually forms a complex with a molecule called GDP, and the cryo-EM studies show that this molecule is released shortly after the RF3•GDP complex enters the ribosome. Once inside the ribosome, RF3 comes into contact with RF1 and with a protein called L12 that is part of the ribosome. A molecule called GTP—which is well known as a source of energy within cells—then binds to RF3, and this causes the shape of the ribosome to change. This change of shape results in the release of RF1 and the formation of a new RF3•GDP complex, which then leaves the ribosome.

Further work is needed to fully understand the role of L12 in these events, but a detailed understanding of the mechanism for terminating the translation of mRNA by the ribosome is coming into view.

Structurally, termination of mRNA translation has been characterized by cryo-EM (*Rawat et al., 2006*) and X-ray crystal (*Petry et al., 2005*; *Laurberg et al., 2008*) structures of A-site bound class-1 RFs interacting with their cognate stop codons. These structures have, in conjunction with molecular dynamics simulations (*Sund et al., 2010*), contributed to present-day understanding of the principles of stop codon recognition by class-1 RFs and of their activation of the peptidyl transfer center for hydrolysis of the ester bond in peptidyl-tRNA. In addition, the X-ray crystal structure of free RF3•GDP (*Gao et al., 2007*) and cryo-EM (*Gao et al., 2007*) and crystal (*Jin et al., 2011*; *Zhou et al., 2012*) structures of post-termination and post class-1 RF release ribosomes in complex with RF3 bound to GTP-analogs are known. The latter complexes indicate large structural changes of RF3 upon GTP binding after release of GDP, and suggest those structural changes to drive the ribosome from its macrostate-1 (MS-I; i.e., characterized by lack of intersubunit rotation; for definition of macrostates, see *Frank et al., 2007*) into a state in which the two ribosomal subunits—30S and 50S—have rotated relative to each other—macrostate-2 (MS-II). Furthermore, these structures of RF3•GDP(N/C)P-bound intersubunit-rotated ribosomes are incompatible with class-1 RF binding (*Gao et al., 2007*; *Jin et al., 2011*; *Zhou et al., 2012*), as previously suggested from biochemical data (*Zavialov and Ehrenberg, 2003*).

In this study we present cryo-EM density maps of the *E. coli* ribosomal post-peptide-release complex (RC; 70S, tRNA$^{fMet}$ and [Met, stop]-mRNA) in association with RF1 alone and in association with both apo-RF3 and RF1. Complexes were obtained at a residual concentration of free GDP and in the absence of GTP. We show that the latter complex contains a novel form of RF1 in close interaction with the apo-form of RF3 and furthermore infer a direct interaction of RF3 with ribosomal protein L7/L12. We use the present data to provide a structural basis for the mechanism of guanine nucleotide exchange on ribosome-bound RF3 and, in conjunction with previous data, to discuss the whole action cycle of RF3.

## Results

### Characterization of the RF3-and-RF1-containing ribosomal release complex

We reconstituted ribosomal RCs in vitro by mixing XR7-Met-Stop (UAA) mRNA-programmed *E. coli* ribosomes containing fMet-tRNA$^{fMet}$ with purified RF1 and apo-RF3 (containing residual guanine nucleotides). To determine the occupancy of RF1 and RF3 in the complex, the reaction was subjected to ultracentrifugation through a sucrose cushion followed by gel analysis in which we estimated the amounts of ribosome-bound RF1 and RF3 by comparing their respective band intensities to band intensities of ribosomal protein S1. The gel showed RF1 and RF3 together on the ribosome in close to 1:1 ratios with S1 in the absence of extra added GDP (*Figure 1A,B*). When the concentration of extra GDP was increased, the RF3 band decreased gradually and virtually disappeared at 200 µM extra GDP concentration, while the RF1 band remained at 1:1 stoichiometry with S1 (*Figure 1A,B*). In a second, similar, experiment performed in the absence of RF1, we detected no binding of RF3 to the ribosome—neither to RC nor to 70S—even with no extra added GDP (*Figure 1C*). From these experiments we could infer that apo-RF3 formed a stable complex with the RF1-bound ribosome at low concentration of free GDP. However, in the absence of RF1 or at elevated GDP concentration, RF3 did not form a stable complex with the ribosome, in line with previous conclusions based on less direct experimental evidence (*Zavialov et al., 2001*, *2002*). Thus, RF3•GDP does not form a stable complex with the RF1-bound ribosome, and we therefore conclude that the reconstituted ribosomal release complex presented in lane 1 of the gel presented in *Figure 1A* contains the apo-form of RF3 along with RF1.

### Reconstruction of RF1- and apo-RF3/RF1-containing RC

The reconstituted RCs were subjected to cryo-EM and single-particle-based reconstruction (*Figure 2—figure supplement 1*) assisted by image classification. Three classes were identified in total. The first class (~14,000 projection images; *Figure 2—figure supplement 2*) yielded a density map of the 70S ribosome in MS-I conformation carrying a P-site tRNA and fragments of an E-site tRNA but with no trace of RF1 or RF3; this class will be disregarded in the following analysis. The second class (~43,000 projection images; *Figure 2A–C*) yielded a density map of the 70S ribosome in MS-I conformation carrying RF1 in the A site, a tRNA in the P site and trace density of a tRNA in the E site; the third class (~29,000 projection images; *Figure 2D–F*) yielded a density map of the 70S ribosome in MS-I conformation carrying apo-RF3, RF1, a P-site tRNA and trace density of an E-site tRNA. These two maps will be referred to as RC-RF1 and RC-RF1•RF3, respectively. RF1 and RF3 were identified by their shapes and positions on the ribosome (*Rawat et al., 2006*; *Gao et al., 2007*, respectively).

In the RC-RF1 map, domain 3 of RF1 reaches into the peptidyl-transferase center of the 50S ribosomal subunit and the super-domain 2/4 of RF1 occupies the decoding center of the 30S ribosomal subunit. In addition, domain 1 of RF1 contacts the N-terminal domain of protein L11 (L11-NTD) and reaches toward the 30S subunit beak (*Figure 2A–C*). In the RC-RF1•RF3 map, domain 3 of apo-RF3 contacts domain 1 of RF1 and extends toward the spur of the 30S subunit (*Figure 2D–F*). Domain 3 of apo-RF3 interacts with 30S protein S12, as observed earlier for RF3 in complex with non-hydrolyzable GTP analogs—GDP(C/N)P—bound to the ribosome in the rotated MS-II form in the absence of a class-1 RF (*Gao et al., 2007*; *Jin et al., 2011*; *Zhou et al., 2012*). However, we observe only a weak interaction between domain 2 of apo-RF3 and 16S rRNA helix 5, whereas the interaction of RF3•GDP(C/N)P with this helix is strong (*Gao et al., 2007*; *Jin et al., 2011*; *Zhou et al., 2012*). Furthermore, we observe domain 1 of RF1 contacting both domain 3 of RF3 and L11-NTD (*Figure 2D–F* and *Figure 2—figure supplement 3B, D*).

Comparison of the RF1 conformation in our RC-RF1•RF3 map (*Figure 2D–F* and *Figure 2—figure supplement 3B, D*) with that in the RC-RF1 map (*Figure 2A–C* and *Figure 2—figure supplement 3A, C*) reveals a large conformational change of RF1 in response to the binding of apo-RF3 resulting in the formation of a bridge from apo-RF3 via RF1 to L11 (*Figure 2—figure supplement 3*). Domain 1 of apo-RF3 is in the vicinity of, but not in contact with, L6/SRL (*Figure 2G,H*, *Figure 3A*). In contrast, a direct contact between RF3 and L6/SRL was observed in the post class-1 RF-release RC-RF3•GDP(C/N)P maps (*Figure 3B*) (*Klaholz et al., 2004*; *Gao et al., 2007*; *Jin et al., 2011*; *Zhou et al., 2012*), with the ribosome in the rotated MS-II state. Hence, in the apo-RF3 and RF1-containing complex, apo-RF3 is docked to the RC framework only through domain 3, while in the RC-RF3•GDP(C/N)P maps

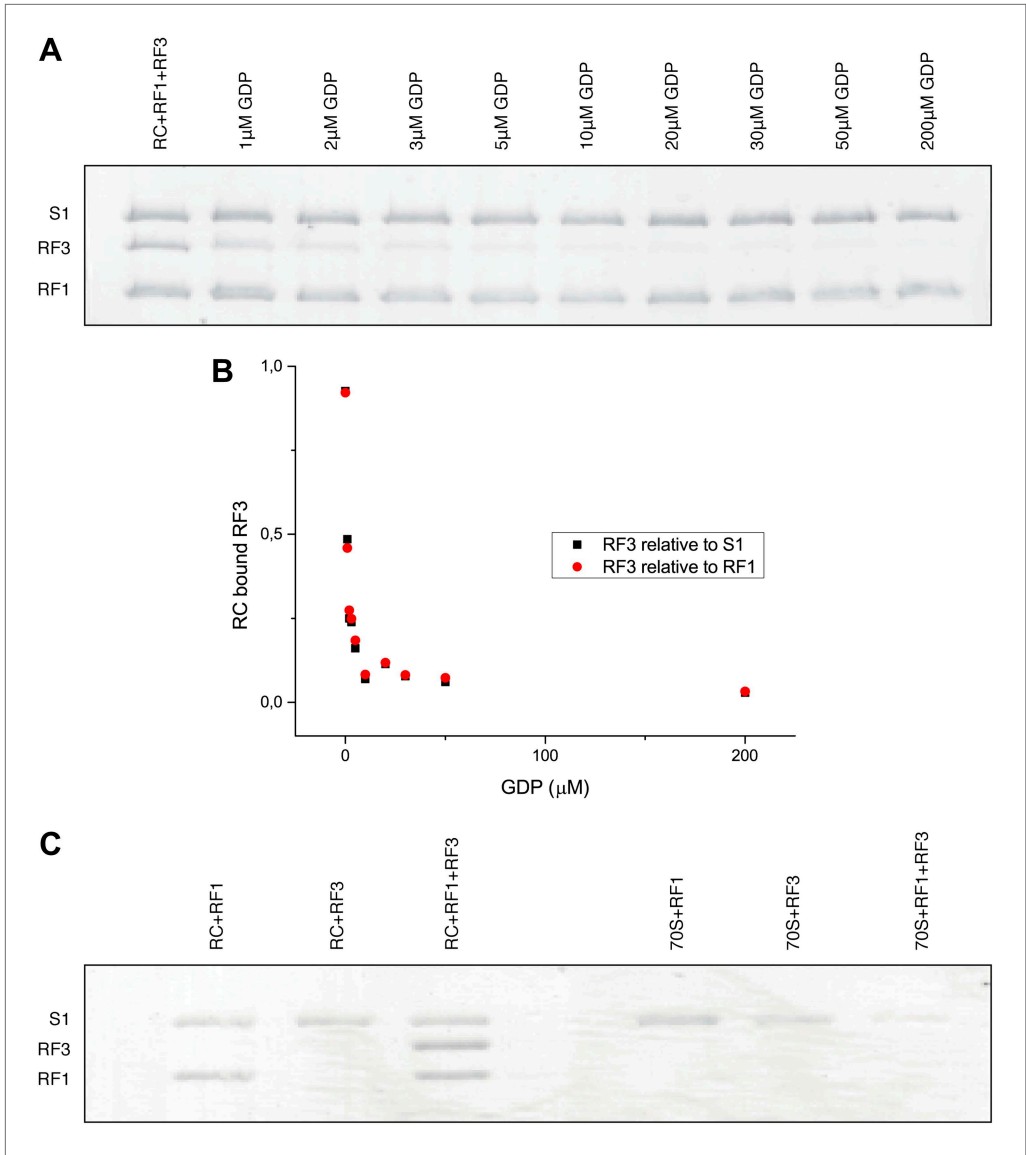

**Figure 1**. Occupancy of RF1 and RF3 in the RC–RF1•RF3 complex and the effect of extra added GDP. (**A**) Titrating GDP into a release complex (RC) programmed with Met-stop mRNA, and containing fMet-tRNA$^{fMet}$ (in the P site), RF1 and RF3. The intensity of the band corresponding to RF3 decreased gradually to zero with increasing GDP concentration, but the S1 and RF1 intensities remained unaltered. (**B**) RF3 band intensity relative to S1 and RF1 plotted vs GDP concentration. (**C**) RC (lanes 1–3) and naked 70S ribosome (lanes 4–6) were incubated with either RF1 alone, RF3 alone or with both RF1 and RF3 as control experiments. RF3 could be detectably bound only to RC—and not to 70S—, when RF1 was present in the complex (lane 3). This confirms that the RF3 bands seen in the gel—in both panels **A** and **C**—indeed arise from functional complex formation.

RF3•GDP(C/N)P is seen to be firmly docked to RC through contacts involving domains 1, 2 and 3 (**Klaholz et al., 2004**; **Gao et al., 2007**; **Jin et al., 2011**; **Zhou et al., 2012**).

While these observations bear out on the interactions between RF1 and RF3 on the ribosome, an additional finding sheds light on ribosomal recruitment of RF3: Domain 1 of RF3 contains an 'arc-like' density (**Figure 2G,H**, **Figure 2—figure supplement 3B, D**) clearly separated from the density of the NTD of L11. An 'arc-like' density was observed in a similar position, but fully bridging RF3 to L11-NTD in the RC-RF3•GDPNP map (**Figure 3B**) (**Gao et al., 2007**). Moreover, a similar bridge to L11-NTD was observed in EF-G- (**Agrawal et al., 1998**, **1999**), EF-Tu- (**Stark et al., 1997**) and IF2-bound ribosome complexes (**Allen et al., 2005**). In the case of EF-G, the bridge was subsequently identified as the CTD

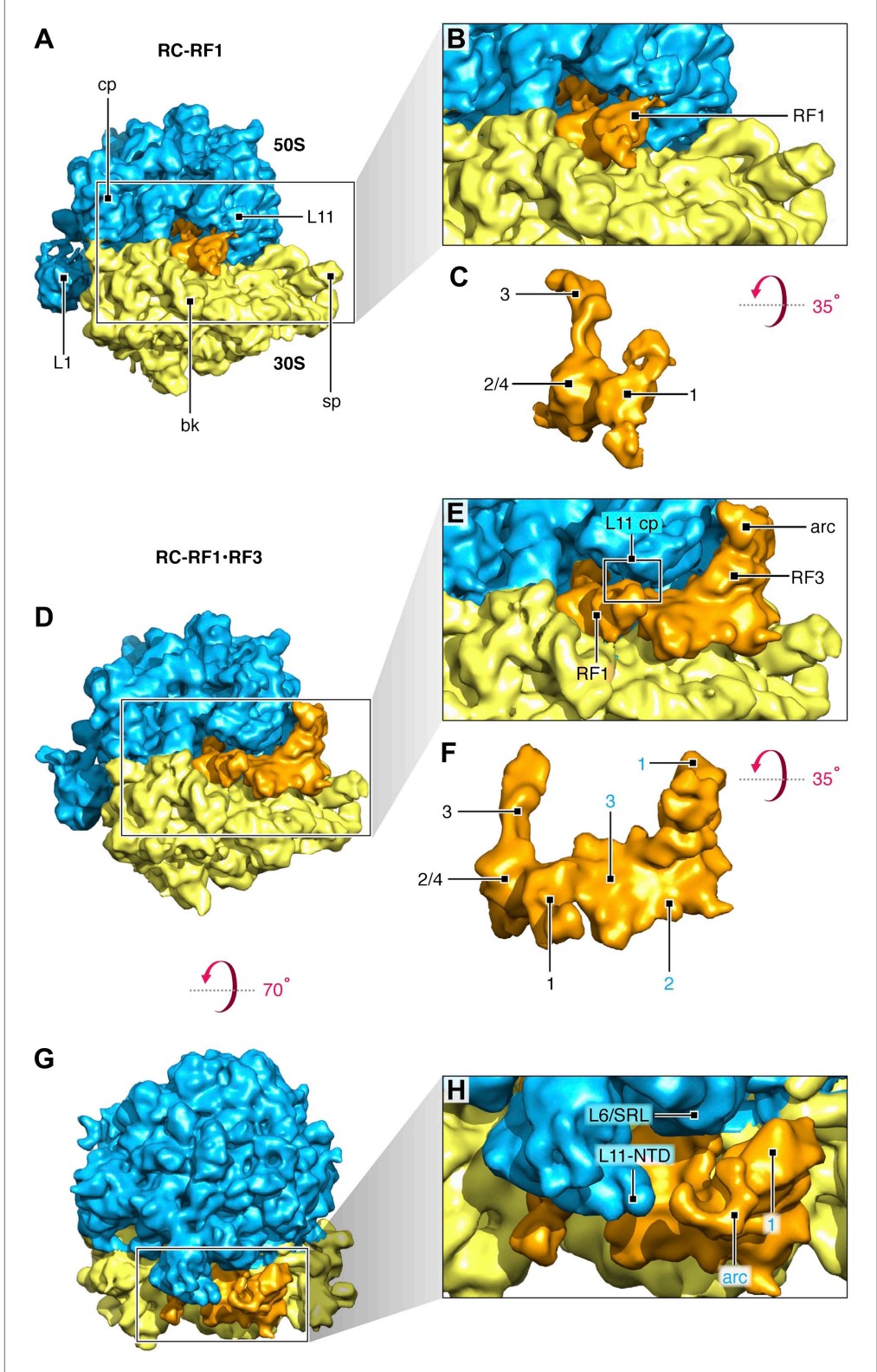

**Figure 2**. Termination complexes in association with RF1 alone and with both RF1 and apo-RF3. (**A**) RC-RF1 (resolution 8.4 Å). (**B**) Close-up of RF1 in the map. (**C**) RF1 density from (**B**). Domains 1, 2/4 and 3 of RF1 are labeled. (**D**) RC–RF1•RF3 (resolution 9.7 Å). (**E**) Close-up of RF1and apo-RF3 in the map. RF1 contacts both L11—L11 contact point is labeled 'L11 cp'— and apo-RF3. (**F**) Density of RF1 and RF3 from (**E**): RF1 domains labeled in black and RF3
*Figure 2. Continued on next page*

*Figure 2. Continued*

domains 1, 2 and 3 in blue. (**G**)–(**H**) In the RC–RF1•RF3 complex, apo-RF3 does not interact with L11 or L6/SRL. Labels: 30S: small ribosomal subunit (yellow), 50S: large ribosomal subunit (blue), cp: central protuberance, L11: ribosomal protein L11, sp: spur, bk: beak, L1: L1 stalk, RF1: release factor 1, RF3: release factor 3, L6/SRL: position of ribosomal protein L6 and the sarcin/ricin loop, L11-NTD: ribosomal protein L11 N-terminal domain, arc: arc-like density. Density maps obtained from the total data set and of the RC-class not occupied by RF1 or RF3 are presented in *Figure 2—figure supplement 1* and *Figure 2—figure supplement 2*, respectively. Additional panels of isolated RF1- and RF1, apo-RF3-densities are presented in *Figure 2—figure supplement 3*. FSC curves for resolution assessment of all density maps obtained can be found in *Figure 2—figure supplement 4*.

The following figure supplements are available for figure 2:

**Figure supplement 1**. Density map resulting from refinement of the full data set.

**Figure supplement 2**. Density map resulting from refinement of the 'RC' class identified in our classification.

**Figure supplement 3**. Segmented RF1 and RF1, apo-RF3 densities.

**Figure supplement 4**. FSC curves for density maps obtained from total data, RC, RC-RF1 and RC-RF1•RF3.

of L7/L12 (henceforth denoted L12; the difference being an N-terminal acetylation in case of L7) (*Datta et al., 2005*). The interaction we observe in the present RC-RF1•RF3 map between L12-CTD and apo-RF3 without involvement of L11-NTD has not previously been reported.

## Interactions involving RF1 and RF3 during termination

To interpret the interactions between the RFs and the ribosome in more detail, we used crystal structures of *E. coli* RF1 (*Graille et al., 2005*; PDB ID 2B3T) and RF3 (*Gao et al., 2007*; PDB ID 2H5E) for modeling in our new RC-RF1 and RC-RF1•RF3 maps and the previous RC-RF3•GDPNP map (*Gao et al., 2007*). Unresolved regions in the structures of both RF1 and RF3 were ab initio- or homology-modeled in accordance with the cryo-EM density maps.

We performed molecular dynamics flexible fitting (MDFF) (*Trabuco et al., 2009*) of our atomic RF1 and RF3 models and the X-ray crystal structure of L12-CTD (*Leijonmarck and Liljas, 1987*; PDB ID 1CTF) to the above-mentioned maps (*Figures 4 and 5*). Root-mean-square deviation plots describing changes in the structures as well as changes in the cross-correlation coefficient between structures and density maps over the course of the MDFF fits are presented in *Figure 4—figure supplement 1*.

In the RC-RF1 map (*Figure 2A*) we observe RF1 in an open conformation bound to the unrotated, MS-I state ribosome (*Figure 4A,C and D*), in accordance with the results of earlier studies of this complex (*Petry et al., 2005*; *Rawat et al., 2006*; *Laurberg et al., 2008*). Moreover, our RC-RF1 map conforms to the position of L11 observed in those studies; a position displaced toward the A site when compared to a 70S ribosome with a vacant A site (*Laurberg et al., 2008*).

By comparing RF1 fitted to our RC-RF1 (*Figure 4A,D*) and RC-RF1•RF3 (*Figure 4B,E*) maps, the conformational change in domain 1 of RF1 upon RF3 binding can be clearly visualized (*Figure 4C*). In our RC-RF1 map, helices α2 and α3 in domain 1 are observed in contact with the P-rich $3_{10}$-helix in L11-NTD (*Figure 4A,D*) (*Petry et al., 2005*; *Rawat et al., 2006*; *Laurberg et al., 2008*). In the RC-RF1•RF3 map, we observe an RF1-mediated structural bridge between L11-NTD and domain 3 of RF3 (*Figure 4E*). The stacking between helices α2 and α3 in domain 1 of RF1 is partly disrupted and helix α2 has undergone a rotation by 72° away from L11 (*Figure 4C,D and E*). Thereby, α2 is positioned within binding distance of domain 3 of RF3 (*Figure 4B,C and E*) while the contact between the tip of α3 and the P-rich $3_{10}$-helix in L11 is maintained (*Figure 4E*) (*Petry et al., 2005*; *Laurberg et al., 2008*).

We compared conformations of apo-RF3 and RF3•GTP fitted to their respective maps with the closed conformation of free RF3•GDP (*Figure 5A–C*). Apo-RF3 in the RC-RF1•RF3 map is observed in a semi-open conformation characterized by a 36° rotation of its domain 3 away from its domain 1 (*Figure 5B,D*). RF3•GTP in the later stage RC-RF3 map is observed in its open conformation with domain 3 rotated 49° away from domain 1 when compared to the closed conformation of free RF3•GDP

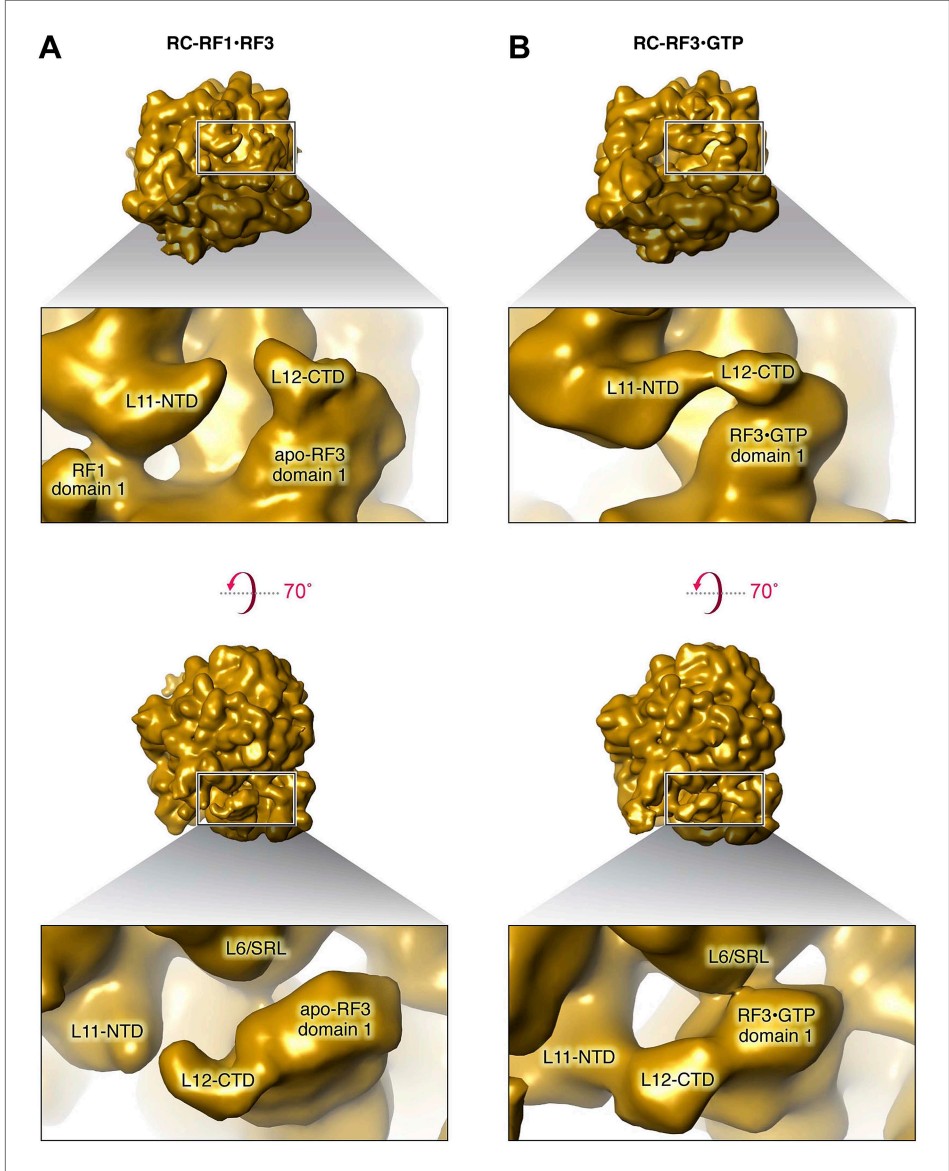

**Figure 3**. Comparison of L11, L6/SRL, arc (L12-CTD) and apo-RF3 in RC-RF1•RF3 and RC-RF3 maps. (**A**) Apo-RF3 is observed in contact with L12-CTD (arc) but L12-CTD is not in contact with L11-NTD. Apo-RF3 domain 1 is not in contact with L6/SRL. In contrast (**B**), RF3•GTP is observed in contact with L6/SRL and in this map (RC-RF3•GTP), L12-CTD is in contact with both RF3 domain 1 and L11-NTD. Both maps are displayed at the resolution of the latter—16 Å—for direct comparison; unsegmented maps were chosen to avoid artifacts on the edges of protein densities and, hence, to present a correct analysis of the interactions occurring.

(*Figure 5C,D*). In the semi-open conformation, amino acid stretches R399-Q411 and N458-N467 of RF3 form a cluster of 12 charged or polar amino acids in two flexible loops of domain 3 at the interface with RF1 (*Figure 5E*). Examining the surface chemistry of helix α2 of RF1 in the region of interaction with RF3, we find 5 charged amino acids (H13, E14, E17, E18 and Q20), which are accessible to interaction with RF3 (*Figure 5F*).

To shed further light on the nature of the binding observed between RF1 and apo-RF3 we examined electrostatic surface potentials in the regions mediating their interaction (*Figure 6A–D*). The surface patch of helix α2 in domain 1 of RF1 that contacts RF3 is negatively charged (*Figure 6A,B*), whereas the complementary surface patch in RF3 is positively charged (*Figure 6C,D*). The combined observations from fitting (*Figure 5D,E*) and surface electrostatics (*Figure 6A–D*) suggest that amino

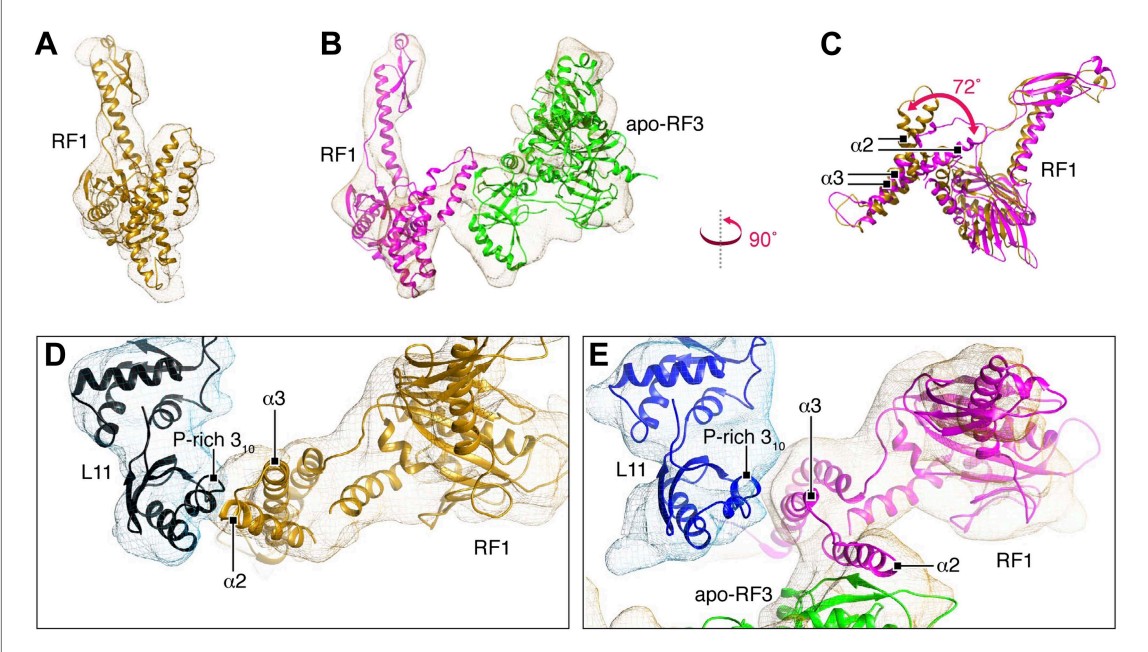

**Figure 4**. Flexible fitting by MDFF focusing on RF1 conformation and interactions. (**A**)–(**C**) Comparison of *E. coli* RF1 conformations after MDFF. (**A**) RF1 in RC-RF1. (**B**) RF1 (and RF3) in RC-RF1•RF3. (**C**) Superimposition of RF1 from (**A**) and (**B**). Arrow in (**C**) denotes the conformational change of domain 1. (**D**) In the RC-RF1 complex, RF1 interacts through helices α2 and α3 with the P-rich $3_{10}$-helix in the L11-NTD. (**E**) In the RC-RF1•RF3 complex α3 (RF1) interacts with the $3_{10}$-helix (L11-NTD), while α2 (RF1) interacts with RF3. α2 and α3 denote helices α2 and α3 in domain 1 of RF1; P-rich $3_{10}$ denotes the P-rich $3_{10}$ helix in L11-NTD. Root mean square deviation and cross correlation coefficient plots for our MD flexible fitting are presented in *Figure 4—figure supplement 1*.

The following figure supplements are available for figure 4:

**Figure supplement 1**. RMSD and CCC values for MD flexible fitting.

acid stretches R399-Q411 and N457-N467 of apo-RF3 are involved in a charge-based interaction with helix α2 of RF1 and residues Q74, E75 and H76 of protein S12 (*Figure 5D*). Thus, binding of apo-RF3 to the ribosomal release complex is obtained by formation of a bridge connecting L11-NTD through domain 1 of RF1 with domain 3 of RF3 (*Figures 2E and 4E*).

As amino acids in these regions of the two release factors are not generally conserved across RF3-expressing species, we investigated whether the electrostatic properties might be conserved (*Figure 6E–G*). Indeed, we find that in X-ray structures available for class-1 RFs from RF3-containing Bacteria (*E. coli* and *S. mutans*; *Figure 6E*), the surface of helix α2 has an overall electronegative surface potential. In addition, examining the surface potential in RF3 from *D. vulgaris* (*Figure 6G*) (*Kihira et al., 2012*) we find a similar positive surface potential in the same position as we observed for RF3 from *E. coli*. In contrast, for Bacteria and Archaea lacking RF3 (*T. thermophilus* and *T. maritima*; *Figure 6F*) the surface potential of helix α2 is overall electroneutral. These observations suggest the existence of charge-based interactions between RF1 and apo-RF3.

## Mutation analysis of the apo-RF3/RF1 interaction

We explored the above-described interaction between apo-RF3 and RF1 in further detail. We generated five RF1 mutants, H13A, E14A, E17A, E18A, and Q20A, and compared the ability of RF3 to recycle these mutants as well as wild-type RF1 during translation termination. In our recycling assay, we mixed RCs containing an MLL tripeptidyl-tRNA in the P site and a UAA stop codon in the A site. The RC was in large excess over RF1, so that many cycles of RF1 action were required to release MLL tripeptides from all RCs. Consistent with earlier reports (*Freistroffer et al., 1997*; *Zavialov et al., 2001*; *Gao et al., 2007*), RF3 increased the recycling rate of wild-type RF1 by a factor of ten in a GTP-dependent manner (*Figure 7*). The ability of RF3 to recycle wild-type RF1 and RF1 mutants E14A,

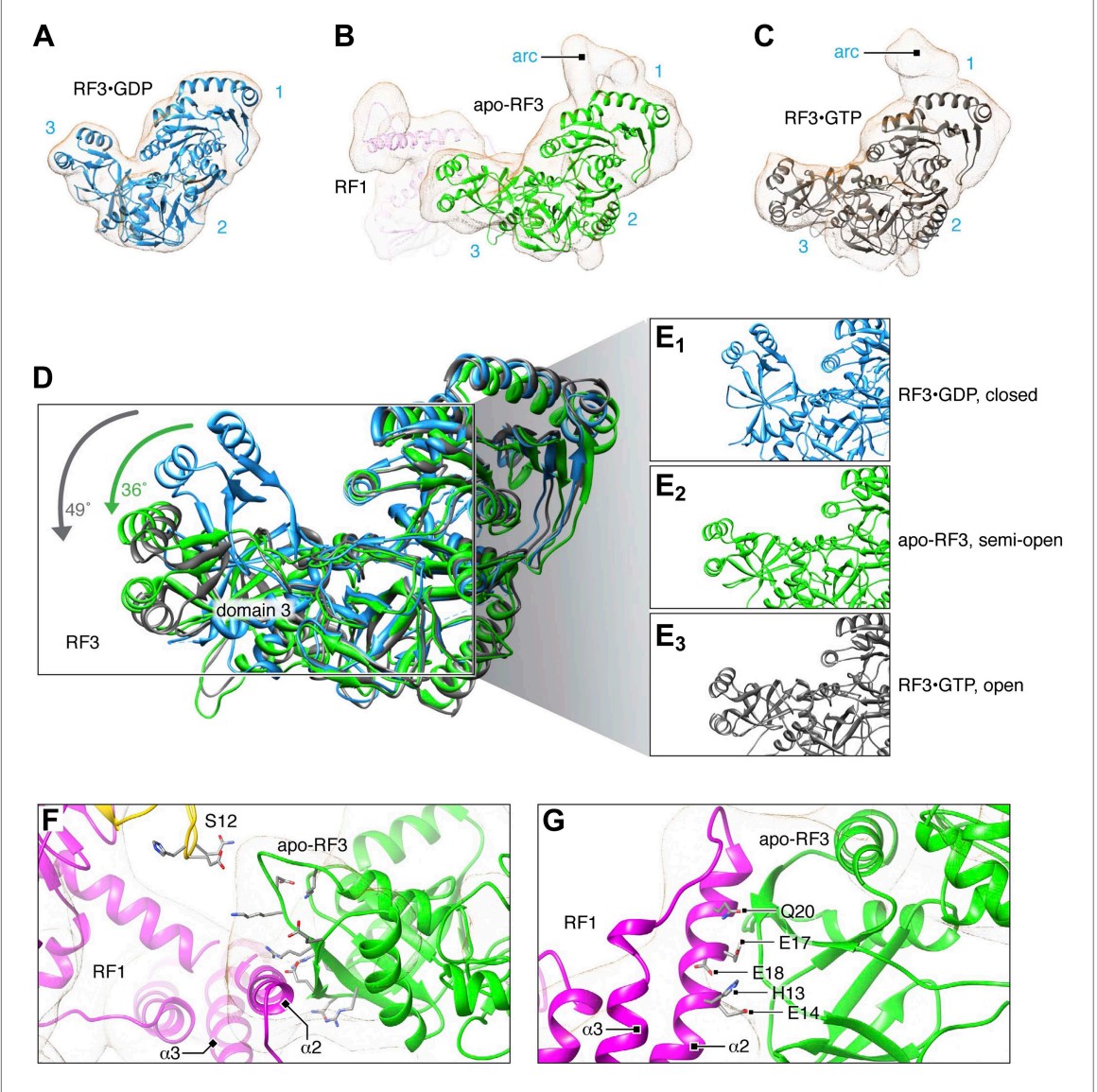

**Figure 5**. Flexible fitting by MDFF focusing on RF3 conformation and interaction with RF1. (**A**)–(**C**) Comparison of *E. coli* RF3 conformations after MDFF. (**A**) Crystal structure of free RF3•GDP displayed in its simulated density. (**B**) apo-RF3 (and RF1) in RC-RF1•RF3 (L12-CTD not displayed) (**C**) RF3 in RC-RF3•GDPNP (***Gao et al., 2007***) (L12-CTD not shown). Superimposition (**D**) of closed conformation RF3•GDP (blue), semi-open apo-RF3 (green) and closed RF3•GTP (gray). Arrows indicate rotations of domain 3. (**E**) Close-ups of domain 3 conformations from (**A**)–(**D**). (**F**) Candidate residues in domain 3 of RF3 for a charge-based interaction with RF1 and 30S protein S12. (**G**) Candidate residues in RF1 helix α2 for a charge-based interaction with RF3. RF3 domains labeled in blue; α2 and α3 denote helices α2 and α3 in domain 1 of RF1. ***Figure 5—figure supplement 1*** shows superimposition of apo-RF3 and RF3•GDPNP.

The following figure supplements are available for figure 5:

**Figure supplement 1**. Movement of domain 1 in RF3.

E17A, and Q20A was similar. In contrast, compared with recycling of wild-type RF1, recycling of the E18A mutant was reduced threefold and recycling of the H13A mutant was reduced 1.4-fold. At the same time, RF3-independent recycling rates of all RF1 variants were indistinguishable, showing that the lower recycling of the E18A and H13A mutants was indeed RF3-mediated.

In our attempt to investigate conservation of the set of amino acids included in our recycling assay, we defined a non-redundant set (***Sethi et al., 2005***) of RF1 sequences from Archaea and Bacteria. For the amino acids E14, E17 and E18, we observed conservation rates—estimated as occurrence

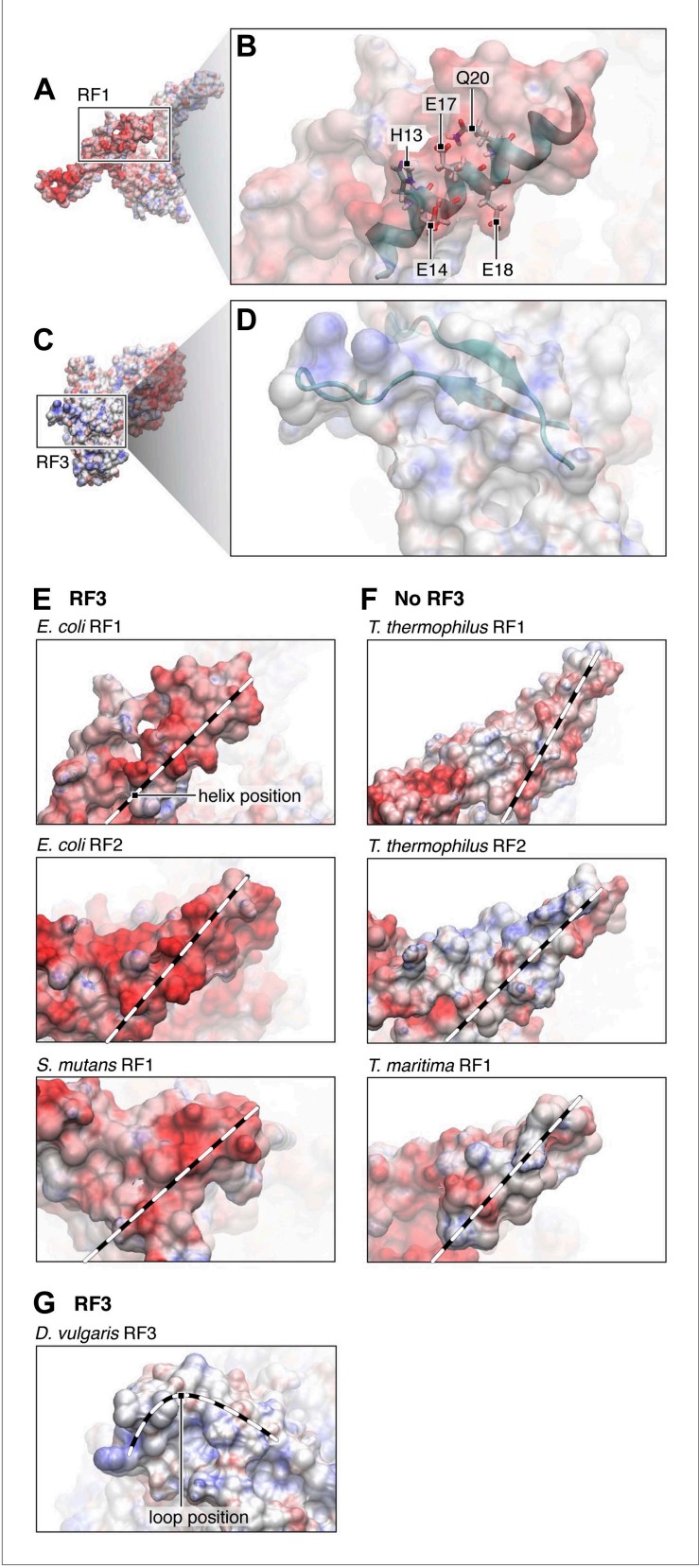

**Figure 6**. Electrostatic surface potentials of RF1 and RF3. (**A**) Electrostatic surface potential of RF1. Red surfaces are electronegative, blue surfaces electropositive. (**B**) Close-up of helix α2 in RF1, where interaction with RF3 is

*Figure 6. Continued on next page*

*Figure 6. Continued*

mediated. Labeled residues (pointers) are candidates (as identified by MDFF) for a direct interaction with RF3. The surface potential of helix α2 is electronegative. (**C**) Electrostatic surface potential of RF3. (**D**) Close-up of the two loops in domain 3 of RF3 posing charged or polar residues for contact with RF1. The surface potential of RF3 in this region is electropositive, pointing to a charge-based interaction between RF1 and RF3. In Bacteria expressing RF3 (**E**) the surface potential of RF1 in the α2 helical region (position of helix α2 indicated by dotted lines) is overall electronegative. In comparison, this region in class-1 RFs from Bacteria and Archaea not expressing RF3 (**F**) is overall electroneutral. (**G**) RF3 from *D. vulgaris* displays an electropositive surface similar to what we observe in *E. coli* RF3 (position of the flexible loop region indicated by dotted line). PDB IDs; *T. thermophilus* RF1: 3D5A, *T. thermophilus* RF2: 2WH3, *T. maritima* RF1: 1RQ0, *E. coli* RF1: current study, modeled/fitted from 2B3T, *E. coli* RF2: 1Gqe, *S. mutans* RF1: 1Zbt, *D. vulgaris* RF3: 3Vqt.

of D or E—of 28%, 40% and 63%, respectively. Q20—no effect in RF3-mediated recycling—and H13 were not conserved. We note that E18—associated with threefold reduction of recycling rate in our assay—is the most conserved amino acid of the candidates.

In light of the findings of surface charge conservation and from the mutation studies, we propose a model in which E18 in the α2-region of RF1 is not only involved in establishing the negatively charged surface patch important for interaction with the positively charged counterpart in RF3 but also involved in a specific interaction with RF3. We attribute the reduction in RF1 recycling rate by the H13A mutation to indirect—steric and/or charge—effects and propose that H13 contributes to optimal orientation of helix α2 including E18 and other residues important for interaction with RF3. However, a direct interaction between H13 and RF3 cannot be excluded.

## RF3 nucleotide loading-and-release

As mentioned above, we observe apo-RF3 in a semi-open conformation. Previously, the conformation of RF3•GDP has been described as closed (*Gao et al., 2007*) and the conformation of RF3•GTP as open (*Gao et al., 2007*; *Jin et al., 2011*; *Zhou et al., 2012*) (*Figure 5A–D*). The apparent dynamics in RF3 conformation lends insight to the mechanism of nucleotide loading-and-release: in the X-ray structure of RF3•GDP, the β-phosphate group of GDP is stabilized by interaction with the conserved H92 of RF3 (*Figure 8B,E*); surprisingly, no $Mg^{2+}$ ion is participating in this stabilization (*Gao et al., 2007*). H92 is part of switch region 2; switch regions 1 and 2 are known to be flexible regions involved in activation and deactivation of GTPase activity in RF3. We note that switch region 1 is disordered in the RF3•GDP crystal structure and therefore not resolved. Looking at the position of H92 in semi-open

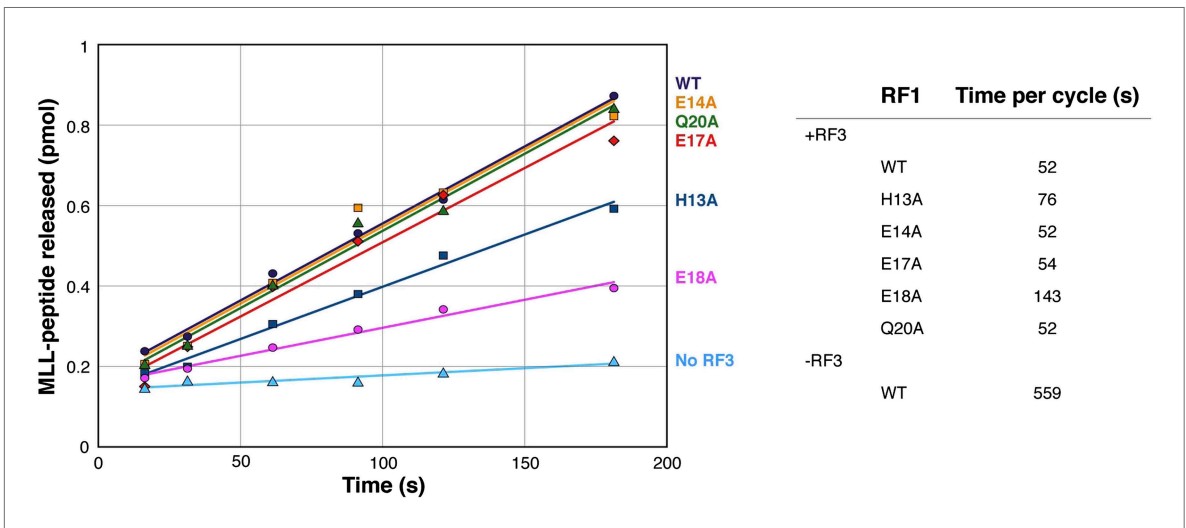

**Figure 7**. MLL release from RC due to RF3-mediated recycling of RF1 (WT and mutants). MLL release from RC due to recycling of RF1 WT and mutants with excess of RF3 over time. The chart shows recycling times for RF1 variants in the presence of RF3.

apo-RF3, we observe H92 retracted from the nucleotide-binding pocket of RF3 (*Figure 8C,E*). In this retracted position—semi-open RF3—H92 is unlikely to participate in nucleotide stabilization. In the later-stage, open RF3•GTP, H92 is observed in a similarly retracted position where it is not participating in stabilization of GTP (*Figure 8D,E*). Moreover, coordination of the β-γ phosphates of GTP is performed by a $Mg^{2+}$ ion, the binding of which is made possible by a well-ordered switch region 1. The $Mg^{2+}$ ion is likely to be essential for GTPase activity (*Zhou et al., 2012*). Hence, the release of GDP from RF3 as it changes conformation from closed (RF3•GDP) to semi-open (apo-RF3) is likely triggered by the retraction of H92 from the nucleotide-binding pocket. This allows for entry of GTP, which is accompanied by a change in RF3-conformation from semi-open (apo-RF3) to open (RF3•GTP) where GTP is stabilized by a $Mg^{2+}$ ion coordinated by a well-ordered switch region 1.

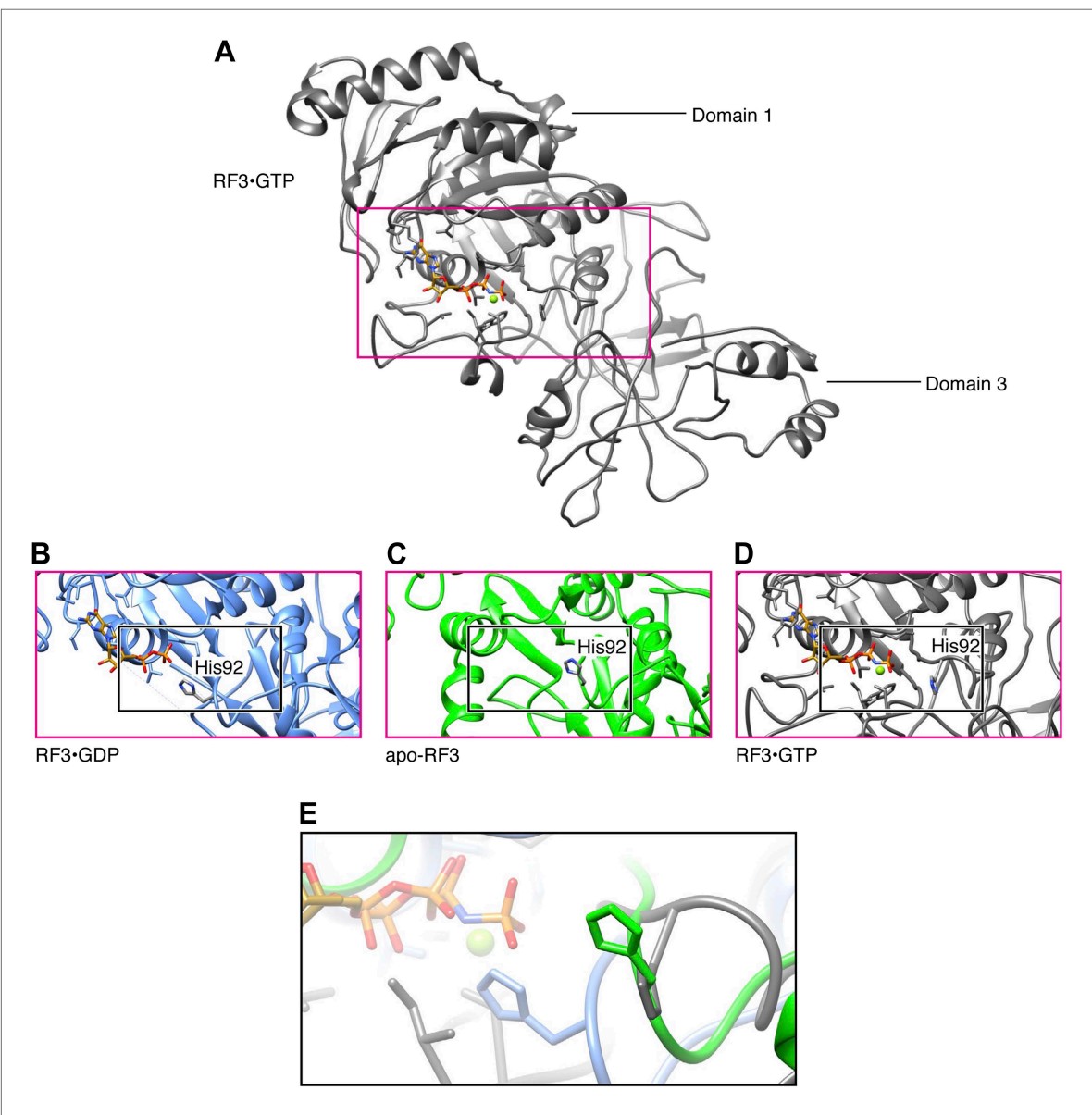

**Figure 8**. RF3 nucleotide loading-and-release. (**A**) RF3•GTP with the nucleotide-binding pocket inside the red box. (**B**)–(**D**) Nucleotide-binding pockets of RF3•GDP, apo-RF3 and RF3•GTP. In RF3•GDP, GDP is stabilized by interaction with H92. In apo-RF3 (**B**) and in RF3•GTP (**C**) H92 is retracted away from the nucleotide-binding pocket. Instead, this stabilization is performed by a $Mg^{2+}$ ion. (**E**) Superimposition of H92 position (black box in [**B**]–[**D**]) showing H92 in its GDP-stabilizing position (blue) and in its retracted positions (green and gray).

## RF3 interacts with L12 during termination

As mentioned above, we observe an 'arc-like' density in contact with domain 1 of apo-RF3 in the RC-RF1•RF3 termination intermediate which we attribute to L12-CTD (*Figure 2D–H*; *Figure 2—figure supplement 3B, D*; *Figure 3* and *Figure 5B*). Based on our fitting, the sole interaction between RF3 and L12 involves helix α7 in the G' subdomain of RF3 and helices α4 and α5 in L12-CTD (*Helgstrand et al., 2007*), while L12-CTD is clearly separated from L11-NTD by a 14-Å gap (*Figure 9A,B* and *Figure 3A*). In contrast, in the map of the later-stage RC-RF3•GDPNP (*Figure 9C,D* and *Figure 3B*) L12-CTD is observed in contact with both L11-NTD and RF3. This suggests that L12-CTD and RF3 interact prior to formation of a structural bridge with L11-NTD. For RF3 to change from apo-RF3 to RF3•GTP, as observed in the RC-RF3•GDPNP complex, domain 3 must rotate even further away from domain 1 (i.e., from 36 to 49°), bringing RF3•GDPNP into its fully open conformation (*Figure 5B–D*). Our fitting predicts that the change from RC-RF1•RF3 to RC-RF3•GDPNP is characterized by a concomitant upward rotation of L12-CTD by ~45° toward the tip of L11-NTD (*Figure 9E*). This rotation is accompanied by a 'hinging-in' movement of helices α4 and α5, whereby α4 rotates inward by ~29° and α5 by ~11° (*Figure 9F*), leading to formation of a structural bridge between domain 1 of RF3 and a loop in L11-NTD (*Figure 9C,D*). Similar flexibility in the α4/α5 region of L12 has been observed in earlier studies (*Leijonmarck and Liljas, 1987*; *Harms et al., 2008*; *Gao et al., 2009*). Along with the change in position and conformation of L12-CTD observed in the RC-RF1•RF3 and RC-RF3•GDPNP complexes, L11 undergoes an upward rotation of ~7°, thereby disrupting its interaction with helix α3 in domain 1 of RF1 (*Figure 9B,D*) and positioning the tip of the L11-NTD for binding to L12-CTD (*Figure 9D*). Furthermore, the ribosome conformation changes—primarily by intersubunit rotation— from MS-I to MS-II, and domain 1 of RF3 moves closer, by a 12° rotation of helix α7, to L6/SRL to form an interaction (*Figure 5—figure supplement 1*) (*Klaholz et al., 2004*; *Gao et al., 2007*; *Jin et al., 2011*; *Zhou et al., 2012*) in accordance with the earlier reported mechanism governing onset of GTPase activity in EF-G (*Clementi et al., 2010*).

To further investigate the nature of L12–CTD interaction with RF3 we examined electrostatic surface properties of both factors in the region of the contact (*Figure 10A–D*). It is evident, that the region in RF3 occupied by helix α7 constitutes a negatively charged surface patch (*Figure 10A,B*), whereas the complementary surface occupied by helices α4 and α5 in L12-CTD is positively charged

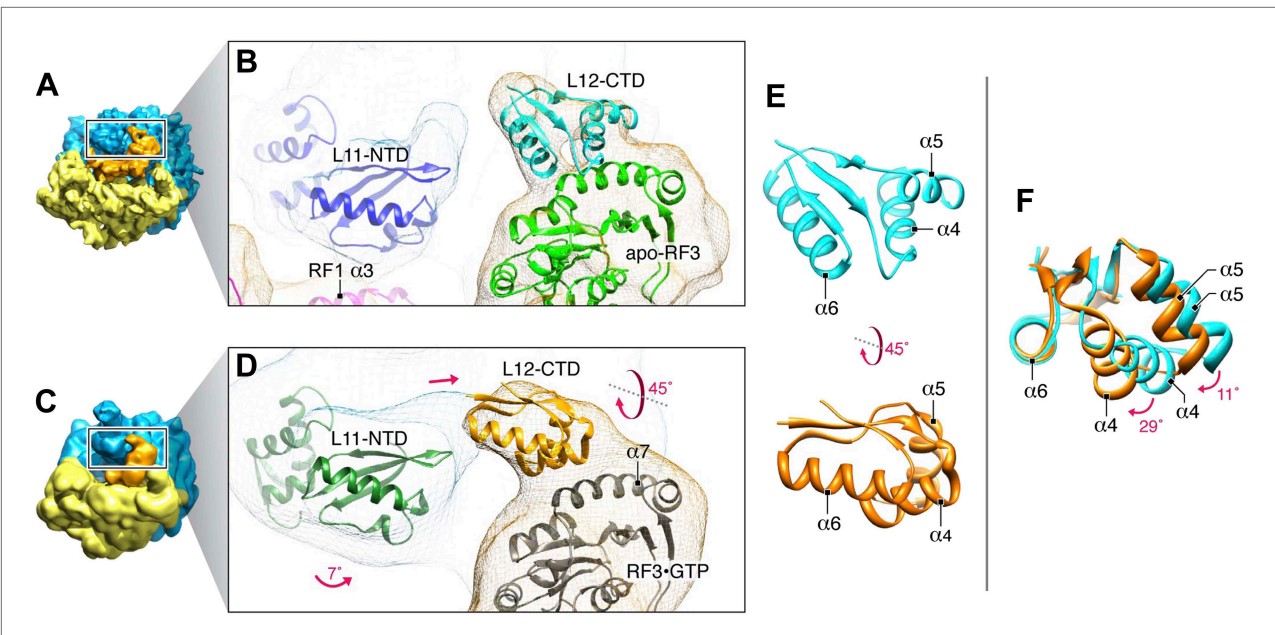

**Figure 9**. Dynamics of interactions involving RF3 and L12-CTD. (**A**) RC-RF1•RF3 map. (**B**) Close-up of RF1 helix α3, apo-RF3, L12-CTD and L11-NTD after fitting. (**C**) RC-RF3•GDPNP map (*Gao et al., 2007*). (**D**) Close-up of RF3•GDPNP, L12-CTD and L11-NTD after fitting. (**E**) L12-CTD position in (**B**) (cyan) and (**D**) (orange) displayed side-by-side. (**F**) Superimposition of L12-CTD from (**B**) and (**D**) showing the 'hinging-in' conformational change observed.

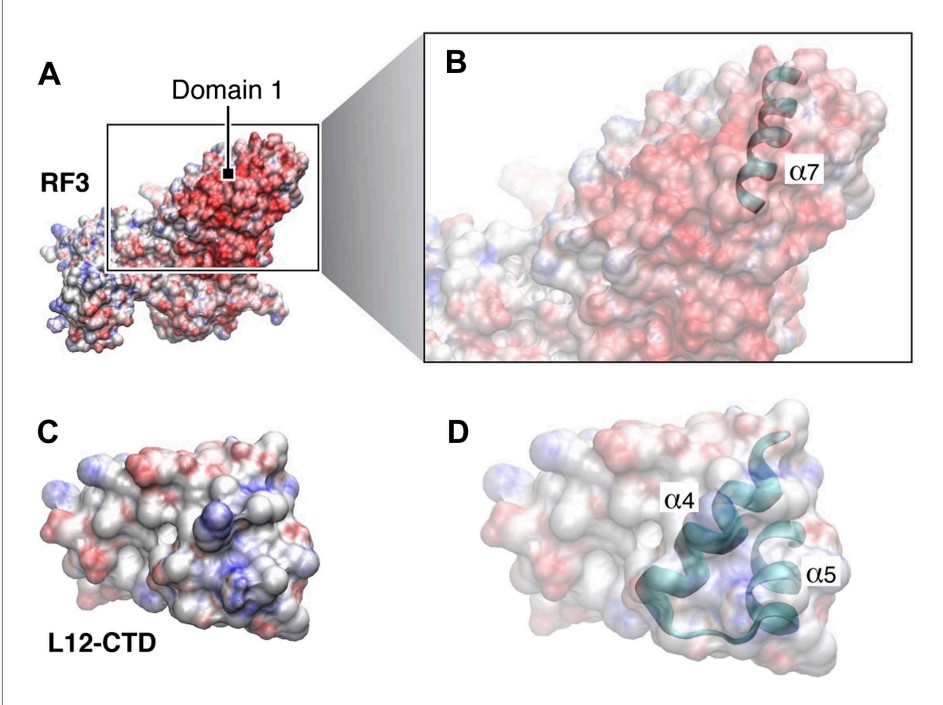

**Figure 10**. Electrostatic surface potentials of L12-CTD, RF3 and other major translational GTPases. Red surfaces are electronegative, blue surfaces electropositive. (**A**) The electrostatic surface potential of RF3 (**B**) Close-up of domain 1 in RF3, displaying the position of helix α7 in the G' subdomain responsible for binding to L12-CTD ([**C**]; close-up in [**D**]). The overall surface potential of the G' subdomain in RF3 is negative and the overall surface charge of helices α4 and α5 in L12-CTD is positive. ***Figure 10—figure supplement 1***: Surface potentials of IF2 (**A**) and (**B**), EF-Tu (**C**) and (**D**), and EF-G (**E**) and (**F**) are all electronegative in domain 1.

The following figure supplements are available for figure 10:

**Figure supplement 1**. Surface potentials of IF1, EF-Tu and EF-G.

---

(***Figure 10C,D***). Examination of electrostatics for the other major translational GTPases (IF2, EF-Tu and EF-G) in the proposed area of interaction shows similar negatively charged patches for potential interaction with L12-CTD (***Figure 10—figure supplement 1A–F***).

## Discussion

The RC-RF1•RF3 structure observed here shows the guanine nucleotide-free apo-form of RF3 in a stable RF1-containing ribosome complex. The existence of such a structure was predicted from biochemical experiments on the action cycle of RF3 (***Zavialov et al., 2001***; ***Sternberg et al., 2009***), but direct experimental evidence has been lacking. The present gel separation analysis proves that indeed it is the apo-form of RF3 in the RC–RF1•RF3 complex. This analysis shows that RF3•GDP does not form a stable complex with the RC, irrespective of whether or not RF1 is present. Furthermore, formation of a stable ribosomal complex with our RF3 preparation requires the presence of RF1 and is possible only when the concentration of free GDP is low. In the chemical equilibrium between the RC–RF1•RF3 complex and the RC–RF1 complex accompanied by free RF3•GDP, the former complex dominates at low concentrations of free GDP, whereas the latter two dominate at high concentrations of free GDP. Hence, from our gel data (***Figure 1***), we infer that the present cryo-EM complex must contain RF1 together with the apo-form of RF3.

It has been shown that the affinity of GTP to free RF3 is at least three orders of magnitude lower than that of GDP and that the average time of spontaneous release of GDP from free RF3 is 30 s (***Zavialov et al., 2001***), a release time that is comparable with the 100 s of GDP from EF-Tu in the absence of EF-Ts (***Ruusala et al., 1982***). The high affinity of GDP to free RF3, possibly due to multiple

contacts between H92 of RF3 and the β-phosphate group of GDP (*Gao et al., 2007*), in combination with the slow dissociation of GDP from RF3 strongly suggests that RF3 exists predominantly in the GDP form in the living cell. Moreover, it has been shown that the class-I RF-bound RC is the guanine nucleotide exchange factor for the G-protein RF3 (*Zavialov et al., 2001*), an observation that suggests that the RC–RF1•RF3 complex contains a form of RF3 from which GDP can readily dissociate and to which GTP can readily associate. That is, the novel complex visualized here is likely to provide the structural basis for the ribosome-dependent guanine nucleotide exchange on RF3. Accordingly, it is RF3 in the compact GDP form that enters the class-1 RF-containing post-termination ribosome complex from which the peptide chain has dissociated after ester bond hydrolysis. As presented here and in previous work (*Zavialov et al., 2001*; *Zavialov et al., 2002*), RF3•GDP has low affinity to the RC. We infer that the semi-open apo-RF3 structure we observe is formed rapidly by its stabilizing contacts with ribosomal protein L12 (*Figure 9*) and a novel form of RF1. The latter, we propose, is formed concomitantly with formation of the apo-form of RF3 (*Figure 11* and *Video 1*). This novel form of RF1 stabilizes the binding of the apo-form of RF3 by contact between helix α2 in RF1-domain 1 and domain 3 of apo-RF3 (*Figure 4E*). The rapid conformational change of RF3 and the stabilization of RF3's apo-form make release of GDP and binding of GTP to RF3 a high-probability event and dissociation of RF3•GDP a low-probability event after initial binding of RF3•GDP to the class-1 RF-containing RC. This implicates L12 and RF1 as essential for the efficient recruitment of RF3 to the RC. L12-CTD has been shown capable of interacting with RF3•GDP (*Helgstrand et al., 2007*) and it is therefore possible that L12 interacts already with free RF3•GDP (*Figure 11*), facilitating the first ribosome-binding step of the factor, but the collection of more experimental data is required to settle this issue.

Conditional on the absence of a peptide chain on the P-site bound tRNA, GTP binding to RF3 induces the open conformation of the factor thereby driving the ribosome into the intersubunit rotated MS-II state (*Klaholz et al., 2004*; *Gao et al., 2007*; *Jin et al., 2011*; *Zhou et al., 2012*). A movement of L11 away from domain 1 of RF1 and toward L12 accompanies this change in ribosome conformation (*Figure 9*). In the process, the structural bridge between the P-rich $3_{10}$-helix in L11, RF1 domain 1 and RF3 domain 3 (*Figures 2E and 4E*) is broken and replaced by a structural bridge formed between the tip of L11-NTD, L12-CTD and domain 3 of RF3•GTP (*Figure 9C,D* and *Figure 3B*), evidently stabilizing RF3•GTP binding to the RC in its MS-II conformation. Concomitantly, RF3•GTP makes contact with L6/SRL on the large ribosomal subunit (*Figure 3* and *Gao et al., 2007*; *Jin et al., 2011*; *Zhou et al. 2012*) As a result of the movement of L11, domain 1 of the class-1 RF is no longer in contact with the ribosome. Thermodynamically, this scenario requires a much lower affinity of GTP to the semi-open than to the

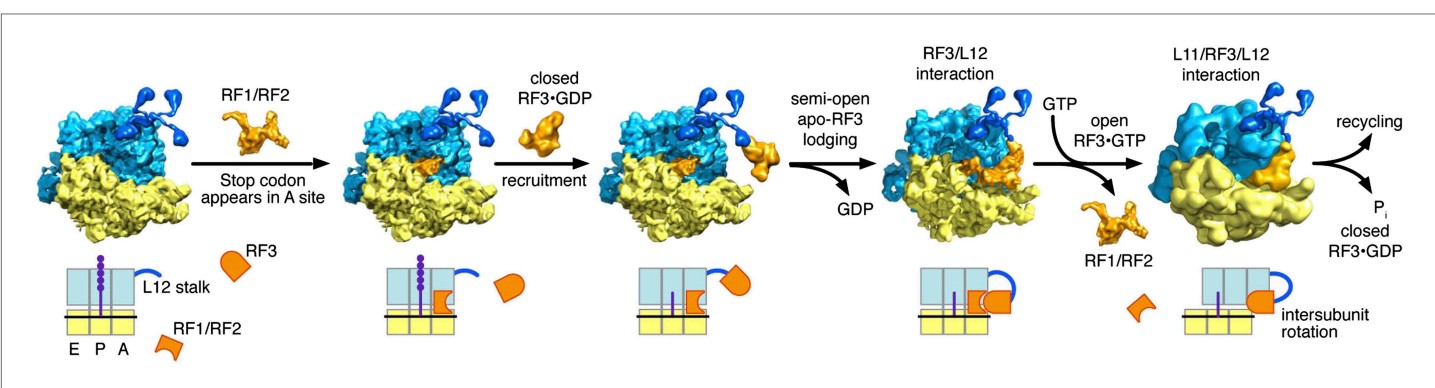

**Figure 11**. Proposed model of translation termination involving intersubunit-, L12-, RF1- and RF3-dynamics. A class-1 RF recognizes its cognate mRNA stop codon in the ribosome and binds in the A site. Here, the class-1 RF mediates release of the nascent protein attached to the P-site tRNA. After nascent protein release, RF3•GDP is recruited to the ribosome. RF3•GDP is in its closed form and does not form a stable complex with the ribosome; it is possible that initital contact between the ribosome and RF3•GDP is mediated by L12-CTD. As RF3 lodges onto the ribosome, GDP is released and apo-RF3 assumes its semi-open conformation. At this point, apo-RF3 is in contact with L12-CTD, the class-1 RF and 30S protein S12. Upon recruitment of GTP to apo-RF3, RF3•GTP assumes its open conformation and the ribosome changes from the unrotated macrostate I to the rotated macrostate II and the class-1 RF leaves the complex. In this state, RF3•GTP is in contact with L6/SRL and L12-CTD, already interacting with RF3, forms a bridge to L11-NTD. This binding state of RF3•GTP marks the onset of RF3's GTPase activity leading to cleavage of GTP and release of Pi. RF3•GDP dissociates from the ribosomal complex in its closed conformation and the ribosome is ready for subunit recycling.

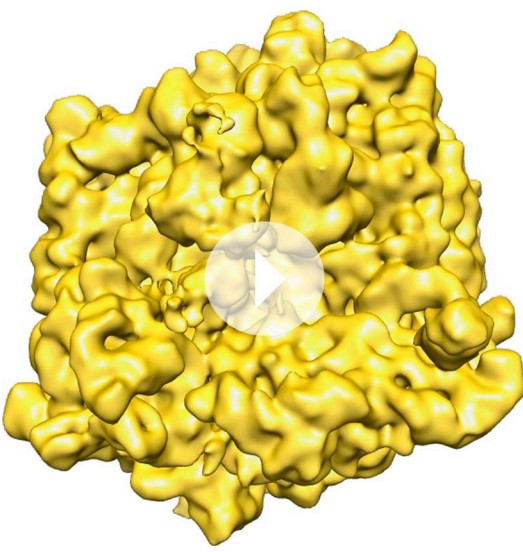

**Video 1**. Animation of proposed model for RF3-mediated termination of translation. RF1 (brown) is positioned in the ribosomal A site in the RC-RF1 complex. Here, its domain 1 makes contact with L11-NTD (black). After initial contact in the cytoplasm between RF3•GDP and L12-CTD, RF3•GDP/L12-CTD lodges onto the ribosome; in this process GDP is released and apo-RF3 adopts its semi-open conformation. RF1 undergoes a conformational change, thereby enabling formation of the stabilizing bridge between L11-NTD, RF1 and apo-RF3 (RC-RF1•RF3; RF1 in magenta, apo-RF3 in green, L12-CTD in cyan). At this point, there is neither a contact between domain 1 of apo-RF3 and L6/SRL nor between L12-CTD and L11. Next, apo-RF3 recruits GTP, thereby changing to its fully open conformation and locking the ribosome in the rotated (MS-II) state. Intersubunit rotation is accompanied by L11 moving away from RF1 and toward RF3•GTP, thereby disrupting its contact with RF1 and triggering the formation of another stabilizing bridge between L11-NTD, L12-CTD and RF3•GTP. Disruption of RF1 interaction with L11 in concert with the ribosome being locked in MS-II is incompatible with continued presence of a class-1 RF in the A site, forcing it to leave the complex (RC-RF3•GTP; L11 in dark green, L12-CTD in orange, RF3•GTP in gray). Moreover, in the rotated RC-RF3•GTP conformation, domain 1 of RF3•GTP contacts ribosomal L6/SRL, marking the onset of GTPase activity. Once GTP is cleaved and $P_i$ is released, RF3•GDP returns to its closed, cytoplasmic state.

open RF3 conformation (**Hauryliuk et al., 2008**), implying that the semi-open conformation of RF3 has low affinity to both GDP and GTP in analogy with the low affinity of both GDP and GTP to the EF–Tu•Ts complex (**Ruusala et al., 1982**). Finally, the intersubunit-rotated MSII state leads to a steric clash between the ribosome and the class-1 RF, which results in the release of the class-1 RF from the A site. RF3 is activated for GTP hydrolysis and rapidly leaves the ribosome in the GDP form.

## Materials and methods

### Translation components
The ribosome (*E. coli*, MRE600), XR-7 mRNAs (Met-stop[UAA] and Met-Leu-Leu-stop[UAA]), fMet-tRNA$^{fMet}$ and the initiation and elongation factors were purified following lab protocols (**Antoun et al., 2004**; **Huang et al., 2010**).

### Purification of the release factors and site-directed mutagenesis of RF1
His-tagged RF1 (kind gift of Valérie Heurgué-Hamard, CNRS, France) and RF3 were purified to single band using His-trap (GEHC) affinity column chromatography. To make RF3 free from extra guanine nucleotides, it was further purified using DEAE-sepharose ion exchange chromatography and gel filtration, by passing through a long superdex-75 column (1.5 × 100 cm) in a diluted solution. RF1 mutants (H13A, E14A, E17A, E18A and Q20A) were created following the standard protocol for QuikChange Mutagenesis (Stratagene, Santa Clara, CA). The mutations were confirmed with DNA sequencing and the mutant proteins were purified with the same protocol as the wild-type.

### RF1 recycling by RF3
For recycling of RF1, an RC containing tripeptidyl 3H-fMet-Leu-Leu tRNA$^{Leu}$ in the P site and a UAA stop codon in the A site was prepared as described in (**Bouakaz et al., 2006**) with minor modifications. The recycling assay was performed with 50 nM RC and 2 nM RF1 (wild-type or mutants) without or with RF3 (0.5 µM) in excess following.

### Assembly of termination complex for cryo-EM
A ribosomal release complex was first made by incubating *E. coli* MRE600 ribosomes (0.5 µM), fMet-tRNA$^{fMet}$ (1 µM) and Met-stop (UAA) mRNA (1 µM) in 1xHEPES-polymix buffer (**Gao et al., 2007**) at 37°C for 10 min. To this, RF1 (2 µM) and RF3 (6 µM) were added and the reaction mix was incubated at 37°C for another 10 min to form the release complex (RC). Occupancy of RF1 and RF3 was checked by running the RC in SDS-PAGE after ultracentrifugation on 30% sucrose cushion (20 mM Tris-HCl, 500 mM NH$_4$Cl, 10.5 mM MgAcetate, 0.5 mM EDTA, 1.1 M Sucrose, 3 mM 2-Mercaptoethanol). Both RF1 and RF3 produced bands with comparable intensity as the ribosomal protein S1 showing

1:1 incorporation in the release complex (*Figure 1*). To study the effect of GDP, similar complexes were formed with increasing concentration of GDP and GTP. Band intensities were digitalized and quantified by the Software UN-SCAN-IT gel TM Version 6.1 (Silk Scientific, Orem, UT).

## Electron microscopy and image processing

To prepare the sample for cryo-EM, the assembled termination complex was diluted in 1xHEPES-polymix buffer to a final concentration of 50 nM. A carbon-coated Quantifoil 2/4 grid (Quantifoil Micro Tools GmbH, Jena, Germany) was prepared following standard cryo procedures (*Grassucci et al., 2007*; *Sengupta et al., 2010*): Grids were glow-discharged for 25 s in an ($H_2$, $O_2$)-atmosphere; 15W and 25s using a plasma cleaner (Gatan Solarus Model 950 Advanced Plasma System; Gatan, Inc., Warrendale, PA) to make them hydrophilic. Samples of 4 μl were pipetted onto carbon-coated grids. Grids were blotted in 100% humidity at 4°C for 5 s and plunge-frozen into liquid ethane cooled by liquid nitrogen using a Vitrobot (FEI, Hillsboro, OR). Images were recorded on SO163 film (Eastman Kodak, Rochester, NY) using an FEI (Eindhoven, The Netherlands) Tecnai Polara operating at 300 kV and a nominal magnification of 59,000× under low-dose conditions (~20 e⁻/Å²) by using the automated data collection system AutoEMation (*Lei and Frank, 2005*). The 798 film micrographs collected were digitized with a step size of 7 μm on an Imaging Scanner (Carl Zeiss, Inc., Jena, Germany). The resulting pixel size was 1.2 Å on the object scale. After visual inspection and evaluation of the micrographs and their power spectra, 643 were selected for further analysis. Particles were chosen via automated particle selection (*Rath and Frank, 2004*) followed by manual verification. The total number of particles used was ~86,000. We employed standard SPIDER scripts in a hierarchical reference-based classification strategy. Initially, the total data set was aligned to a density map of the 70S ribosome in complex with a P-site and an E-site tRNA (70S-P, E) filtered to 20 Å and refined on a course angular grid over several rounds of refinement according to standard SPIDER protocols (*Frank et al., 1996*; *Shaikh et al., 2008*) which include a 3D projection alignment procedure with correction of the contrast transfer function (CTF) using 82 defocus groups, with defocus varying from −1.2 to −4.6 μm. Next, the projection images were classified on the basis of cross-correlation coefficients with two references, namely (70S-P, E) and an RF1-containing ribosomal complex having P- and E-site tRNAs (EMDB ID 1184; *Rawat et al., 2006*). Two classes were identified (70S-P, E: ~14,000 projection images, and EMDB ID 1184: ~72,000 projection images). The initial orientation assignments to the (70S-P, E) map were chosen as starting point for refinement of each class. The RF1-containing class was then classified further with the RF1-containing map (EMDB ID 1184) as first reference and a second reference ('RF1/RF3-chimera') generated by adding the segmented RF3 density from the RF3•GDPNP map (EMDB ID 1302; *Gao et al., 2007*) to the RF1-containing map (EMDB ID 1184). Two subclasses of projection images (EMDB ID 1184: ~43,000 projection images, and RF1/RF3-chimera: ~29,000 projection images) were identified and refined separately as described above. Following convergence of standard refinement, a second refinement routine was performed in which CTF correction was performed at the 2D projection-level. CTF-corrected projection images were then pooled together and a 3D reconstruction was obtained. Only a subset of the best-aligned particles (based on cross-correlation with the reference in each view) was used for reconstruction until the very last rounds of small-angle refinement. The final resolutions for the maps were 8.0 Å for the total data set (*Figure 2—figure supplement 1*), 11.8 Å (RC; *Figure 2—figure supplement 2*), 8.4 Å (RC-RF1) and 9.7 Å (RC-RF1•RF3), using a Fourier shell correlation (FSC) cutoff of 0.5 (*Figure2—figure supplement 4*).

## Modeling RF1 and RF3

Models of *E. coli* RF1 and RF3 were built based on their crystal structures (*Graille et al., 2005*; *Gao et al., 2007*) (PDB ID: 2B3T and 2H5E, respectively).

RF1 was modeled in two conformations fitting the cryo-EM maps of RC-RF1 and RC-RF1•RF3. Missing and unstructured segments, in the crystal structure, as well as segments interacting with the mRNA, and the P-site tRNA, were modeled mostly by homology and in part manually. Residues 70–105, 227–241 and 285–306 were modeled by homology to the *T. thermophilus* RF1 crystal structure in complex with the 70S ribosome (*Laurberg et al., 2008*) (PDBID: 3D5A), using SWISS-MODEL (*Schwede et al., 2003*; *Arnold et al., 2006*; *Bordoli et al., 2009*). Residues 57–69 were modeled manually using PyMOL (The Pymol Molecular graphics system, Version 1.3, Schrödinger, LLC) to position them in the main chain and then Swiss-Pdb Viewer (*Guex and Peitsch, 1997*) to modify their

torsions in order to fit the density map in a region between helices 2 and 3 that obviously presents different structure from the *T. thermophilus* RF1 crystal structure.

RF3 was modeled interacting with RF1 in the *E. coli* 70S ribosome by modifying the conformation of RF3 first to roughly fit the corresponding cryo-EM density map. The fit was realized using UCSF Chimera (*Pettersen et al., 2004*) by fitting domains I, II and III of the crystal structure as three independent rigid bodies into the RF3 density in the RC-RF1•RF3 map. Subsequently, Swiss-Pdb Viewer was used to accommodate the coils linking different fitted domains. Residues 37–67, missing from the crystal structure, were modeled de novo in SWISS-MODEL using SYMPRED for secondary structure prediction.

## Molecular dynamics flexible fitting

Using VMD (*Humphrey et al., 1996*), three systems were prepared for MDFF. One system consisted of RF1, *E. coli* ribosomal proteins L11, L16 and S12, P-site tRNA, mRNA, the rRNA surrounding scaffold (16S rRNA residues 10–25, 502–542, 787–797, 885–928, 950–1070, 1190–1232, 1390–1421, and 1479–1525 and 23S rRNA residues 1052–1108, 1903–1969, 2051–2067, and 2445–2615) and surrounding magnesium cations. The coordinates for the system, except for RF1, P-site tRNA and mRNA, were taken from the *E. coli* 70S crystal structure by *Berk et al. (2006)* (PDBID: 2I2P and 2I2T). P-site tRNA and mRNA coordinates were taken from the *T. thermophilus* crystal structure of the 70S ribosome in complex with RF1 (*Laurberg et al., 2008*) (PDBID: 3D5A). A second system was created similar to the system mentioned above, the only difference being the addition of RF3 and the CTD of L7/L12 (*Leijonmarck and Liljas, 1987*; PDB ID 1CTF). The third system consisted of RF3, the CTD of L7/L12, *E. coli* ribosomal proteins L11 and S12, as well as rRNA segments scaffolding these two proteins (16S residues 10–25, 502–542, 885–892, 907–927, 1390–1418 and 1482–1508 and 23S residues 1052–1108) (*Berk et al., 2006*).

All systems were embedded in a box of TIP3P water molecules with an extra 12 Å padding in each direction. The system was neutralized by potassium ions and an excess of KCl was added to ~ 0.2 M. All the simulated systems were prepared using CHARMM force field parameters (Combined CHARMM All-Hydrogen Topology File for CHARMM22 Proteins and CHARMM27 Lipids [*MacKerell et al., 1998*]). Each system was relaxed by 8000 minimization steps in NAMD (*Phillips et al., 2005*). After relaxation, an additional potential was added to the systems based on cryo-EM segments comprising only the molecules to be simulated, and MDFF simulations were performed. Cryo-EM segments were prepared using the SeggerR plug-in in UCSF Chimera (*Pintilie et al., 2010*). Simulations of RF1- and RF1•RF3-containing complexes were run for 600 ps, whereas the simulation of the RF3-containing complex was terminated after 1 ns. The RF1/RF3 interface in the simulations of RF1•RF3-containing complexes was fine-tuned manually, using PyMOL, in accordance to the outcome of the mutagenesis assays performed. Angular changes between RF3-conformations were estimated based on alignment of amino acid residues 97–105, 121–133 and 328–343.

## APBS calculation

Poisson-Boltzmann electrostatic properties for RF1, RF2 and RF3 models were calculated with the Adaptive Poisson-Boltzman Solver (*Baker et al., 2001*; *Dolinsky et al., 2004*). Potentials range from −15.91V to 12.62V and the results were displayed in VMD using an RWB color scale ranging from −14 to 14.

## Building of the model shown in Figure 11

For the model presented in *Figure 11*, we merged density maps of RC-RF1, RC-RF1•RF3 and RC-RF3 with simulated density maps of L12. The L12 maps were rendered at a resolution of 11 Å using the NMR structure of isolated L12 (PDB ID 1RQU) and the threshold for display was chosen with emphasis on linker visibility.

## Animation

An animation script was generated and '*Video 1*' was rendered in UCSF Chimera.

## Acknowledgements

We thank Robert A Grassucci for assistance with the microscopy, Melissa Thomas for assistance with the preparation of the illustrations and Harry Kao, PhD for assistance in computer hardware issues.

# Additional information

## Funding

| Funder | Grant reference number | Author |
|---|---|---|
| Howard Hughes Medical Institute | | Jesper Pallesen, Yaser Hashem, Joachim Frank |
| National Institutes of Health | R01 GM29169 | Jesper Pallesen, Yaser Hashem, Joachim Frank |
| Swedish Research Council | 2010-2619 | Gürkan Korkmaz, Ravi Kiran Koripella, Chenhui Huang, Måns Ehrenberg, Suparna Sanyal |
| Knut and Alice Wallenberg Foundation for RiboCORE | KAW2011.0081 | Gürkan Korkmaz, Ravi Kiran Koripella, Chenhui Huang, Måns Ehrenberg, Suparna Sanyal |
| Carl Tryggers Foundation | CTS 10:330 | Gürkan Korkmaz, Ravi Kiran Koripella, Chenhui Huang, Suparna Sanyal |
| Wenner Gren Foundation | | Gürkan Korkmaz, Ravi Kiran Koripella, Chenhui Huang, Suparna Sanyal |
| Swedish Research Council | 2011-6008 | Gürkan Korkmaz, Ravi Kiran Koripella, Chenhui Huang, Måns Ehrenberg, Suparna Sanyal |
| Swedish Research Council | 2008-6593 | Gürkan Korkmaz, Ravi Kiran Koripella, Chenhui Huang, Måns Ehrenberg, Suparna Sanyal |

The funders had no role in study design, data collection and interpretation, or the decision to submit the work for publication.

## Author contributions

JP, Conception and design, Acquisition of data, Analysis and interpretation of data, Drafting or revising the article, Contributed unpublished essential data or reagents; YH, ME, SS, JF, Conception and design, Analysis and interpretation of data, Drafting or revising the article; GK, RKK, CH, Acquisition of data, Analysis and interpretation of data

# Additional files

## Major datasets

The following previously published datasets were used:

| Author(s) | Year | Dataset title | Dataset ID and/or URL | Database, license, and accessibility information |
|---|---|---|---|---|
| Rawat U, Gao H, Zavialov A, Gursky R, Ehrenberg M, Frank J | 2006 | Interactions of the release factor RF1 with the ribosome as revealed by cryo-EM | 1184; http://www.ebi.ac.uk/pdbe/entry/EMD-1184 | Publicly available at EMBL-EBI (http://www.ebi.ac.uk/) |
| Gao H, Zhou Z, Rawat U, Huang C, Bouakaz L, Wang C, Liu Y, Zavialov A, Gursky R, Sanyal S, Ehrenberg M, Frank J, Song H | 2007 | RF3 induces ribosomal conformational changes responsible for dissociation of class I release factors | 1302; http://www.ebi.ac.uk/pdbe/entry/EMD-1302 | Publicly available at EMBL-EBI (http://www.ebi.ac.uk/) |
| Graille M, Heurgue-Hamard V, Champ S, Mora L, Scrima N, Ulryck N, van Tilbeurgh H, Buckingham RH | 2005 | Molecular basis for bacterial class 1 release factor methylation by PrmC | 2B3T; http://www.rcsb.org/pdb/explore/explore.do?structureId=2b3t | Publicly available at the RCSB Protein Data Bank (http://www.rcsb.org/) |
| Gao H, Zhou Z, Rawat U, Huang C, Bouakaz L, Wang C, Cheng Z, Liu Y, Zavialov A, Gursky R, Sanyal S, Ehrenberg M, Frank J, Song H | 2007 | Crystal structure of E.coli polypeptide release factor RF3 | 2H5E; http://www.rcsb.org/pdb/explore/explore.do?structureId=2h5e | Publicly available at the RCSB Protein Data Bank (http://www.rcsb.org/) |

| | | | | |
|---|---|---|---|---|
| Berk V, Zhang W, Pai RD, Cate JHD | 2006 | Crystal Structure of Ribosome with messenger RNA and the Anticodon stem-loop of P-site tRNA | 2I2P; http://www.rcsb.org/pdb/explore/explore.do?structureId=2I2P | Publicly available at the RCSB Protein Data Bank (http://www.rcsb.org/) |
| Berk V, Zhang W, Pai RD, Cate JHD | 2006 | Crystal Structure of Ribosome with messenger RNA and the Anticodon stem-loop of P-site tRNA | 2I2T; http://www.rcsb.org/pdb/explore.do?structureId=2I2T | Publicly available at the RCSB Protein Data Bank (http://www.rcsb.org/) |
| Laurberg M, Asahara H, Korostelev A, Zhu J, Trakhanov S, Noller HF | 2008 | Structural basis for translation termination on the 70S ribosome | 3D5A; http://www.rcsb.org/pdb/explore/explore.do?structureId=3d5a | Publicly available at the RCSB Protein Data Bank (http://www.rcsb.org/) |
| Leijonmarck M, Liljas A | 1987 | Structure of the C-terminal domain of the ribosomal protein L7/L12 from Escherichia coli at 1.7 A | 1CTF; http://www.rcsb.org/pdb/explore/explore.do?structureId=1ctf | Publicly available at the RCSB Protein Data Bank (http://www.rcsb.org/) |
| Bocharov EV, Sobol AG, Pavlov KV, Korzhnev DM, Jaravine VA, Gudkov AT, Arseniev AS | 2004 | NMR structure of L7 dimer from E.coli | 1RQU; http://www.rcsb.org/pdb/explore.do?structureId=1rqu | Publicly available at the RCSB Protein Data Bank (http://www.rcsb.org/) |
| Weixlbaumer A, Jin H, Neubauer C, Voorhees RM, Petry S, Kelley AC, Ramakrishnan V | 2008 | Insights into translational termination from the structure of RF2 bound to the ribosome | 2WH3; http://www.rcsb.org/pdb/explore.do?structureId=2WH3 | Publicly available at the RCSB Protein Data Bank (http://www.rcsb.org/) |
| Shin DH, Brandsen J, Jancarik J, Yokota H, Kim R, Kim SH | 2004 | Crystal structure of peptide releasing factor 1 | 1RQ0; http://www.rcsb.org/pdb/explore/explore.do?structureId=1rq0 | Publicly available at the RCSB Protein Data Bank (http://www.rcsb.org/) |
| Vestergaard B, Van L, Andersen G, Nyborg J, Buckingham R, Kjeldgaard M | 2001 | Polypeptide chain release factor 2 (RF2) from Escherichia coli | 1GQE; http://www.rcsb.org/pdb/explore/explore.do?structureId=1gqe | Publicly available at the RCSB Protein Data Bank (http://www.rcsb.org/) |
| Joint Center for Structural Genomics | 2005 | Crystal structure of Peptide chain release factor 1 (RF-1) (SMU.1085) from Streptococcus mutans at 2.34 A resolution | 1ZBT; http://www.rcsb.org/pdb/explore/explore.do?structureId=1zbt | Publicly available at the RCSB Protein Data Bank (http://www.rcsb.org/) |
| Kihira K, Shimizu Y, Shomura Y, Shibata N, Kitamura M, Nakagawa A, Ueda T, Ochi K, Higuchi Y | 2012 | Crystal structure analysis of the translation factor RF3 | 3VQT; http://www.rcsb.org/pdb/explore/explore.do?structureId=3vqt | Publicly available at the RCSB Protein Data Bank (http://www.rcsb.org/) |
| Allen GS, Zavialov A, Gursky R, Ehrenberg M, Frank J | 2005 | IF2, IF1, and tRNA fitted to cryo-EM data of E. COLI 70S initiation complex | 1ZO1; http://www.rcsb.org/pdb/explore/explore.do?structureId=1zo1 | Publicly available at the RCSB Protein Data Bank (http://www.rcsb.org/) |
| Abel K, Yoder MD, Hilgenfeld R, Jurnak F | 1996 | Whole, unmodified, EF-Tu (elongation factor Tu) | 1DG1; http://www.rcsb.org/pdb/explore/explore.do?structureId=1dg1 | Publicly available at the RCSB Protein Data Bank (http://www.rcsb.org/) |
| Gao Y-G, Selmer M, Dunham CM, Weixlbaumer A, Kelley AC, Ramakrishnan V | 2009 | The structure of the ribosome with elongation factor G trapped in the post-translocational state (part 3 of 4) | 2WRK; http://www.rcsb.org/pdb/explore/explore.do?structureId=2wrk | Publicly available at the RCSB Protein Data Bank (http://www.rcsb.org/) |

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
