## [Decision Letter]

Thank you for choosing to send your work entitled “RF3 action is Regulated by a Class-I Release Factor and L7/L12 during Translation Termination” for consideration at *eLife*. Your article has been reviewed in depth by a Senior editor and 2 other reviewers.

The Senior editor and the other reviewers discussed their comments and the Senior editor has assembled the following comments.

The release of peptides from the ribosome is triggered by two classes of ribosome release factors working together. Class I release factors (e.g., RF1) recognize the stop codon and catalyze cleavage of the peptide from the last tRNA. Class II release factors (e.g., RF3) facilitate the release of Class I release factors from the ribosome, in a GDP/GTP dependent manner. This paper presents the results of past and present cryo-EM reconstructions, aided by molecular dynamics, of the release factors RF1 and RF3 bound to ribosomal complexes.

There are two distinct new reconstructions, one with just RF1 bound to the ribosome, and one with both RF1 and RF3 (both complexes contain a peptidyl-tRNA bound in the P site with a stop codon poised in the A site). The analysis involves the visualization of fewer than 100,000 images, 43,000 corresponding to RF1 only bound and 29,000 corresponding to RF1 and apo-RF3 bound to the ribosome; both of these complexes are unrotated (MS-I or classical). These structures reveal some new features of interactions between these factors and specific features of the ribosome (in particular proteins L12 and L11) and with one another. These new reconstructions are compared to crystal structures and EM reconstructions obtained previously for ribosomal complexes with RF3 loaded with GTP analogs.

By providing the first views of a bacterial ribosome bound to both RF1 and RF3, this paper fills an important gap in the structural understanding of translation termination. In particular, the analysis identifies conformational changes in both RF1 and RF3 induced by their mutual presence on the ribosome, and also identifies points of contact with the ribosome and between RF1 and RF3, as well as how these contacts change through the cycle. There are, however, some limitations in the analysis that need to be addressed by the authors before the paper can be considered for publication by *eLife*.

*1) Nature of the interactions with RF3*.

The understanding in the field, largely based on biochemistry from the Ehrenberg lab, argues that the ribosome is the guanine nucleotide exchange factor (GEF) for RF3. In the model, RF3:GDP binds to the ribosome following peptide release catalyzed by RF1/2, the ribosome facilitates exchange of GDP for GTP, and RF3:GTP binds to the rotated state of the ribosome (MS-II or hybrid); this state of the ribosome leads to release of RF1/2 as a result of steric clashes. There is structural data corresponding to some of these steps – RF3 off the ribosome bound to GDP and RF3:GDPNP on the ribosome (indeed trapped in MS-II). That said, the field has been unable to observe the binding of RF3:GDP to the ribosome using any method. Here, the authors get around this problem by using apo-RF3, which apparently does bind to the MS-I state.

1.1) An issue concerns the identification of the nucleotide-loading state of RF3 in the analysis. The Ehrenburg group had identified the ribosome as the guanine nucleotide exchange factor for RF3 (Zavialov, Buckingham, Ehrenberg, Cell, 107:2001). As shown in that study, GDP forms a high affinity complex with RF3, with a nanomolar dissociation constant and a very slow off-rate. In the present study, samples are prepared by simply running RF3 through a gel filtration column. As noted by Zavialov et al., such a procedure will most likely not release nucleotide from RF3. When the ribosome is added to nucleotide-loaded RF3 in the presence of excess unbound nucleotide, Zavialov et al. note that bound nucleotide is rapidly exchanged. In the present study, however, excess nucleotide is not present, and it is not clear that the complex formed is actually free of GDP. While this ambiguity does not change the nature of the reconstructions, it does alter the possible interpretations concerning the actual complex that is visualized. The authors need to either provide citations to existing evidence that their sample preparation procedure removes nucleotide from RF3, or demonstrate that it is removed. Alternatively, the interpretation of the results has to be stated more carefully.

1.2) Another concern is that the authors spend most of the Discussion rationalizing how these data easily fit the prevailing model (“our study … leads to a comprehensive model for class-1 RF release…”) rather than considering other possibilities – i.e., that RF3 does not bind to the ribosome in its GDP form. The Discussion fits these snapshots into a putative biochemical cycle, indicating in very strong terms how the various snapshots fit together and how the missing gaps are filled. It is recommended that the authors more clearly state what is known and how their data either potentially fit into the proposed cycle of termination, or how their data might be differently interpreted given the fact that RF3:GDP does not readily bind.

*2) The resolution of the analysis and the limits it places on mechanistic interpretations*.

2.1) One of the new findings of the paper concerns the dynamic interactions of RF3 with the L12-CTD. These come from comparison of two maps at different resolutions, where the main point is that there is a protein–protein contact in the lower resolution map (RC-RF3 bound to GDPNP (a structure from 5 years ago at 15 Å)) that is not present in a higher resolution one (RC-RF1-RF3 at 9 Å). The apparent additional connectivity in the former may simply be due to the lower resolution. It would have been more appealing if the authors had compared the two states in maps filtered to a common resolution (the poorer one), and then see if differences are still obvious. Or, even better, if they presented an improved map of the RC-RF3GDPNP.

2.2) Given that the ribosome is implicated as the exchange factor for RF3, the present analysis provides little in the way of a molecular understanding of how this works. Various states of RF3 such as “closed”, “semi-open” and “open” are identified in the paper, but the manuscript does not make clear how these relate to nucleotide loading and release. It could be that the limited resolution precludes such interpretation, but the authors should provide some guidance and possible mechanism for nucleotide release.

---

## [Author Response]

1) Nature of the interactions with RF3.

*The understanding in the field, largely based on biochemistry from the Ehrenberg lab, argues that the ribosome is the guanine nucleotide exchange factor (GEF) for RF3. In the model, RF3:GDP binds to the ribosome following peptide release catalyzed by RF1/2, the ribosome facilitates exchange of GDP for GTP, and RF3:GTP binds to the rotated state of the ribosome (MS-II or hybrid); this state of the ribosome leads to release of RF1/2 as a result of steric clashes. There is structural data corresponding to some of these steps – RF3 off the ribosome bound to GDP and RF3:GDPNP on the ribosome (indeed trapped in MS-II). That said, the field has been unable to observe the binding of RF3:GDP to the ribosome using any method. Here, the authors get around this problem by using apo-RF3, which apparently does bind to the MS-I state*.

A ribosomal complex with apo-RF3 is important in its own right, since it provides the key to understanding guanine nucleotide exchange on ribosome-bound RF3. We have, in other words, solved a different and, in our opinion, more interesting problem than to find a substitute for a ribosome complex with RF3•GDP. Moreover, the reviewers acknowledge this point themselves: “By providing the first views of a bacterial ribosome bound to both RF1 and RF3 this paper fills an important gap in the structural understanding of translation termination.”

*1.1) An issue concerns the identification of the nucleotide-loading state of RF3 in the analysis. The Ehrenburg group had identified the ribosome as the guanine nucleotide exchange factor for RF3 (Zavialov, Buckingham, Ehrenberg, Cell, 107:2001). As shown in that study, GDP forms a high affinity complex with RF3, with a nanomolar dissociation constant and a very slow off-rate. In the present study, samples are prepared by simply running RF3 through a gel filtration column. As noted by Zavialov et al., such a procedure will most likely not release nucleotide from RF3. When the ribosome is added to nucleotide-loaded RF3 in the presence of excess unbound nucleotide, Zavialov et al. note that bound nucleotide is rapidly exchanged. In the present study, however, excess nucleotide is not present, and it is not clear that the complex formed is actually free of GDP. While this ambiguity does not change the nature of the reconstructions, it does alter the possible interpretations concerning the actual complex that is visualized. The authors need to either provide citations to existing evidence that their sample preparation procedure removes nucleotide from RF3, or demonstrate that it is removed. Alternatively, the interpretation of the results has to be stated more carefully*.

As the reviewers point out, there is some GDP present in the RF3 preparations. Due to the gel-filtration step during RF3 preparation, the concentration of total GDP is at most at 1:1 stoichiometry with the concentration of active RF3. To unambiguously resolve the question of RF3 nucleotide loading-state in our cryo-EM density map with RF1 and RF3 on the ribosome, we have now performed a series of control experiments.

First, aliquots of RC, RF1, and purified RF3 were incubated with additionally added GDP ranging from 0 to 200 μM. The mixtures were subsequently subjected to ultracentrifugation through sucrose cushions and the resulting pellets were analyzed by gel electrophoresis to describe their contents of RF1, RF3, and ribosomal protein S1. With no additionally added GDP, the pellets contained RF1, RF3, and S1 at a stoichiometry of ∼1:1:1. By incrementally increasing the amounts of additionally added GDP to a concentration of 50 μM, the pelleted samples contained unchanging amounts of RF1 and S1, whereas they contained decreasing amounts of RF3; RF3 concentration eventually decreased to zero. In addition, we detected no binding of RF3 to the ribosome — RC as well as 70S — in the absence of RF1 even with no additionally added GDP.

Taken together, these experiments demonstrate that under conditions similar to those chosen for cryo-EM-complex formation, GDP-bound RF3 could not bind stably to the RF1-bound RC. Only at a very low concentration of free GDP was RF3 able to form a ribosomal complex, conditional on the presence of a complex of RC and RF1. The inference here is that only the apo-form of RF3 could form a strong complex with RC and that this complex formation was conditional on the presence of RF1. This means that *at a sufficiently low concentration of free GDP, the strong binding of apo-RF3 to RC outcompetes the strong binding of free GDP to free RF3.* Therefore, it is not necessary to completely remove GDP from RF3 in solution to obtain a stable complex of apo-RF3 and the post-termination ribosomal RC. However, when the concentration of free GDP is increased, the free RF3•GDP complex is favored increasingly relative to the RC-bound apo-form of RF3. In terms of thermodynamic equilibrium, the fraction of RF3 in ribosome bound apo-form is given by:[RF1·RC·RF3][RF1·RC·RF3]+[RF3·GDP]=11+[GDP][RF1·RC]·KRCKGDP

Here, [RF1·RC·RF3] is the concentration of apo-RF3 in the ribosomal complex with RF1, [RF3·GDP] is the concentration of free RF3 in complex with GDP, [RF1·RC] is the concentration of ribosomal RC in complex with RF1 only, [GDP] is the concentration of free GDP, KGDP is the dissociation constant for binding of free GDP to free apo-RF3, andKRC is the dissociation constant for binding of free apo-RF3 to the RF1·RC complex. In our titration with GDP we shifted the fraction of apo-RF3 in the ribosomal complex from a value near one ([GDP][RF1·RC]·KRCKGDP<<1) to a value close to zero ([GDP][RF1·RC]·KRCKGDP>>1). The expression shows that the ribosome-bound fraction of apo-RF3 depends not only on the concentration of GDP and the small dissociation constantKGDP, but also on the concentration of RF1·RC and the — likewise small — dissociation constantKRC. These control experiments reveal that what is observed in the cryo-EM complex is, indeed, apo-RF3 in contact with RF1 in a strong — small KRC — ribosomal release complex. They confirm previous conclusions ([51]; [53]; Moras et al. 2005), based on less direct biochemical data, and they are now presented in the revised version of our manuscript (body and Figure 1).

*1.2) Another concern is that the authors spend most of the Discussion rationalizing how these data easily fit the prevailing model (“our study … leads to a comprehensive model for class-1 RF release…”) rather than considering other possibilities – i.e., that RF3 does not bind to the ribosome in its GDP form. The Discussion fits these snapshots into a putative biochemical cycle, indicating in very strong terms how the various snapshots fit together and how the missing gaps are filled. It is recommended that the authors more clearly state what is known and how their data either potentially fit into the proposed cycle of termination, or how their data might be differently interpreted given the fact that RF3:GDP does not readily bind*.

In the previous paragraph we described direct experimental evidence that RF3 in the apo-form readily forms a stable complex with the ribosome in the presence of a small concentration of free GDP. This experimental condition was chosen to make it possible to study the functionally important ribosomal complex with RF1 and apo-RF3. In the living cell, the situation is different. Here, GTP is available at a high concentration and GDP at a lower, but significant concentration. The experimental evidence attesting to the fact that RF3 in the GDP-form binds to the post-termination ribosome *in vivo* and in *in vivo* mimicking biochemical experiments is very strong. It is based on the very high affinity of GDP to RF3 compared to that of GTP, on the very small rate of exchange of GDP on free RF3, on the rapid rate of exchange of ribosome bound RF3•GDP in the presence of RF1 or RF2, on the comparatively rapid turnover of RF3 *in vivo*, and on the non-negligible concentration of GDP in the living bacterial cell ([51]; [53]; Moras *et al.* 2005). To this we may add that these previously published pieces of evidence and conclusions have withstood more than ten years of scrutiny by our colleagues.

At the same time, however, the reviewers have a valid point in asking for a lucid summary of previously obtained experimental evidence. We have now, rather than solely relying on our previously published experiments in highly renowned journals, composed a more informative text regarding the very important issue of how to properly place the present and previous structures of free RF3 and of RF3 in ribosomal complexes in the complete action cycle of RF3.

2) The resolution of the analysis and the limits it places on mechanistic interpretations.

*2.1) One of the new findings of the paper concerns the dynamic interactions of RF3 with the L12-CTD. These come from comparison of two maps at different resolutions, where the main point is that there is a protein–protein contact in the lower resolution map (RC-RF3 bound to GDPNP (a structure from 5 years ago at 15 Å)) that is not present in a higher resolution one (RC-RF1-RF3 at 9 Å). The apparent additional connectivity in the former may simply be due to the lower resolution. It would have been more appealing if the authors had compared the two states in maps filtered to a common resolution (the poorer one), and then see if differences are still obvious. Or, even better, if they presented an improved map of the RC-RF3GDPNP*.

We are providing the requested side-by-side view (Figure 3) in which the regions of the two maps in question are displayed side-by-side, and from two different angles. Both maps are now displayed at the common, lower resolution. As is readily visible, the conclusions presented in the manuscript regarding RF3-interaction with L6/SRL, L11, and L12-CTD are confirmed under these conditions.

*2.2) Given that the ribosome is implicated as the exchange factor for RF3, the present analysis provides little in the way of a molecular understanding of how this works. Various states of RF3 such as “closed”, “semi-open” and “open” are identified in the paper, but the manuscript does not make clear how these relate to nucleotide loading and release. It could be that the limited resolution precludes such interpretation, but the authors should provide some guidance and possible mechanism for nucleotide release*.

Upon the reviewers’ request we have now given a tentative analysis (body and Figure 8) describing a possible mechanism of nucleotide loading-and-release. The analysis relies on several premises:

1) It is known from the X-ray structure of free RF3•GDP that switch region 1 is disordered (16).

2) In the structure of free RF3•GDP (Figure 8, shown in blue), switch region 2 is well ordered; His92 — part of switch region 2 and conserved — makes several contacts with the β-phosphate group of GDP (16).

3) The RF3 H92A P90Q mutant shows a 2–3-fold increase in dissociation equilibrium constant and in dissociation rate constant compared to WT RF3 (16).

4) In the RC-RF1•RF3 map, apo-RF3 (Figure 8, shown in green) is observed in a ‘semi-open’ conformation. Here, switch region 2 has moved away from the position observed in free RF3•GDP resulting in a displacement of His92 from its previous position by ∼7.6 Å. In this position, His92 is unlikely to participate in coordination of a β-phosphate group of GDP.

5) In the X-ray structure of RF3•GTP (Figure 8, shown in gray) on the ribosome, RF3 is observed in its fully ‘open’ conformation. Here, His92 is observed in a similar retracted position as in apo-RF3. The distance between the position of His92 in RF3•GTP and RF3•GDP is ∼5.7 Å and His92 is not involved in GTP-binding (54).

6) In RF3•GTP, switch region 1 is well ordered and partakes in stabilization of GTP as well as in coordination of a Mg^2+^ ion. This Mg^2+^ ion is stabilizing the γ-phosphate of GTP — interacting at the β-γ linkage — and the Mg^2+^ ion is likely to be essential for GTPase activity (54).

Based on these observations, a possible mechanism for nucleotide loading-and-release could be as follows: a condition that must be met in order for RF3 to bind GDP with high affinity is the coordination of GDP’s β-phosphate group by His92. In the ‘closed’ conformation of RF3•GDP, switch region 2 is positioning His92 favorably for such coordination. As RF3 binds to the ‘canonical’ ribosomal GTPase-binding site — interacting with 16S rRNA helix 5 and protein S12 — on the ribosome, it concomitantly changes conformation to ‘semi-open’. Here, the position of switch region 2 results in His92 being significantly retracted from the nucleotide-binding pocket. GDP dissociates rapidly as a consequence of this conformational change leading to the stable complex of RC-RF1•RF3 we are reporting. Binding of GTP to RF3 correlates with the ordering of switch region 1 and coordination of the γ-phosphate of GTP through interaction with a Mg^2+^ ion, which is essential for GTPase activity. In the process of GTP acquisition, intersubunit rotation takes place, resulting in interaction between RF3 and ribosomal L6/SRL (54). Moreover, a stabilizing connection of RF3 to L11-NTD is established through the bridging performed by L12-CTD.